# ELBO-ING STEIN MIXTURES

## ABSTRACT

Stein variational gradient descent (SVGD) (Liu & Wang, 2016) is a particle-based technique for Bayesian inference. SVGD has recently gained popularity because it combines the ability of variational inference to handle tall data with the modeling power of non-parametric inference. Unfortunately, variance estimation scales inversely with the dimensionality of the model leading to underestimation, meaning more particles are required to represent high-dimensional models adequately. Stein mixtures (Nalisnick & Smyth, 2017) alleviate the exponential growth in particles by letting each particle parameterize a distribution. However, the inference algorithm proposed by Nalisnick & Smyth (2017) can be numerically unstable. We show that their algorithm corresponds to inference with the Rényi $\alpha$-divergence for $\alpha = 0$ and that using other values for $\alpha$ can lead to a more stable inference. We empirically study the performance of Stein mixtures inferred with different $\alpha$ values on various real-world problems, demonstrating significantly and consistently improved results when using $\alpha = 1$, which corresponds to using the evidence lower bound (ELBO). We call this instance of our algorithm ELBO-within-Stein. An easy-to-use version of the inference algorithm (for arbitrary $\alpha \in \mathbb{R}$) is available in the deep probabilistic programming language NumPyro (Phan et al., 2019).

## 1 INTRODUCTION

The ability of Bayesian deep learning to quantify the uncertainty of predictions by deep models is causing a surge of interest in using these techniques (Izmailov et al., 2021). Bayesian inference aims to describe i.i.d. data $\mathcal{D} = \{\mathbf{x}_i\}_{i=1}^n$ using a model with latent a variable $\mathbf{z}$. Bayesian inference does this by computing a posterior distribution $p(\mathbf{z}|\mathcal{D})$ over the latent variable given a model describing the joint distribution $p(\mathbf{z}, \mathcal{D}) = p(\mathcal{D}|\mathbf{z})p(\mathbf{z})$. We obtain the posterior by following Bayes' theorem,

$$p(\mathbf{z}|\mathcal{D}) = \Pi_{i=1}^n p(\mathbf{x}_i|\mathbf{z})p(\mathbf{z})/p(\mathcal{D}),$$

where $p(\mathcal{D}) = \int_{\mathbf{z}} \prod_{i=1}^n p(\mathbf{x}_i|\mathbf{z})p(\mathbf{z})d\mathbf{z}$ is the normalization constant. For most practical models, the normalization constant lacks an analytic solution or poses a computability problem, complicating the Bayesian inference problem.

Stein variational gradient descent (SVGD) (Liu & Wang, 2016) is a recent technique for Bayesian inference that uses a set of particles $\mathcal{Z} = \{\mathbf{z}_i\}_{i=1}^N$ to approximate the posterior $p(\mathbf{z}|\mathcal{D})$. The idea behind SVGD is to iteratively transport $\mathcal{Z}$ according to a force field $S_\mathcal{Z}$, called the Stein force. The Stein force is given by

$$S_\mathcal{Z}(\mathbf{z}_i) = \mathbb{E}_{\mathbf{z}_j \sim q_\mathcal{Z}}[k(\mathbf{z}_i, \mathbf{z}_j)\nabla_{\mathbf{z}_j} \log p(\mathbf{z}_j|\mathcal{D}) + \nabla_{\mathbf{z}_j} k(\mathbf{z}_i, \mathbf{z}_j)], \tag{1}$$

where $k(\cdot, \cdot)$ is a reproducing kernel (Berlinet & Thomas-Agnan, 2011), $q_\mathcal{Z} = N^{-1}\sum_i \delta_{\mathbf{z}_i}$ is the empirical measure on the set of particles $\mathcal{Z}$, $\delta_\mathbf{x}(\mathbf{y})$ represents the Dirac delta measure, which is equal to 1 if $\mathbf{x} = \mathbf{y}$ and 0 otherwise, and $\nabla_{\mathbf{z}_j} \log p(\mathbf{z}_j|\mathcal{D})$ is the gradient of the posterior with respect to the $j$-th particle. The technique is scalable to tall data (i.e. datasets with many data points) and offers the flexibility and scope of techniques such as Markov chain Monte Carlo (MCMC). SVGD is good at capturing multi-modality (Liu & Wang, 2016; Wang & Liu, 2019), and has useful theoretical interpretations such as a set of particles following a gradient flow (Liu, 2017) or in terms of the properties of kernels (Liu & Wang, 2018).

The main problem is that SVGD suffers from the curse of dimensionality: variance estimation scales inversely with dimensionality (Ba et al., 2021). Nalisnick & Smyth (2017) suggest resolving this by

using a Stein mixture (SM). SMs lift each particle to the parameters of a variational distribution $q$, also called a guide. The idea is that each guide in the Stein mixture represents the density of multiple particles in SVGD, thereby reducing the number of particles needed to represent a posterior. The Nalisnick & Smyth algorithm introduces guides by replacing each posterior gradient $\nabla_{\mathbf{z}_j} \log p(\mathbf{z}_j|\mathcal{D})$ in Equation (1) with the corresponding gradient of the marginal log-variational likelihood given by

$$\log p(\mathcal{D}|\boldsymbol{\phi}_j) = \log \mathbb{E}_{q(\mathbf{z}|\mathcal{D},\boldsymbol{\phi}_j)} \left[ \frac{p(\mathcal{D}, \mathbf{z})}{q(\mathbf{z}|\mathcal{D}, \boldsymbol{\phi}_j)} \right]. \tag{2}$$

Here, we denote the particles by $\Phi = \{\boldsymbol{\phi}_j\}_{i=1}^N$ instead of $\mathcal{Z} = \{\mathbf{z}_i\}_{i=1}^N$ to emphasize they parameterize guide components $q(\mathbf{z}|\boldsymbol{\phi}_i, \mathcal{D})$. The change in gradient corresponds to minimizing $D_{\mathrm{KL}}[q_\Phi(\boldsymbol{\phi}) \parallel p(\boldsymbol{\phi}|\mathcal{D})]$ rather than $D_{\mathrm{KL}}[q_{\mathcal{Z}}(\mathbf{z}) \parallel p(\mathbf{z}|\mathcal{D})]$, as in SVGD. Note that the line between the model $p$ and guide $q$ becomes blurred, as $p(\mathcal{D}|\boldsymbol{\phi})$ is random in both data ($\mathcal{D}$), as is usually the case, but also in the guide hyper-parameters $\boldsymbol{\phi}$ (Ranganath et al., 2016; Nalisnick & Smyth, 2017). To distinguish the two we subsequently refer to $p(\mathcal{D})$ as the evidence and $p(\mathcal{D}|\boldsymbol{\phi})$ as the hierarchical likelihood.

The Stein force using the log hierarchical likelihood, which we call the hierarchical Stein force $S_\Phi^{\mathrm{H}}$, becomes

$$S_\Phi^{\mathrm{H}}(\boldsymbol{\phi}_i) = \mathbb{E}_{\boldsymbol{\phi}_j \sim q_\Phi} \left[ k(\boldsymbol{\phi}_i, \boldsymbol{\phi}_j) \nabla_{\boldsymbol{\phi}_j} \log \mathbb{E}_{q(\mathbf{z}|\mathcal{D},\boldsymbol{\phi}_j)} \left[ \frac{p(\mathcal{D}, \mathbf{z})}{q(\mathbf{z}|\mathcal{D}, \boldsymbol{\phi}_j)} \right] + \nabla_{\boldsymbol{\phi}_j} k(\boldsymbol{\phi}_i, \boldsymbol{\phi}_j) \right], \tag{3}$$

where $q_\Phi$ is an empirical measure analogous to $q_{\mathcal{Z}}$.

Inference converges (i.e. reaches a fixed point) when $S_\Phi^{\mathrm{H}}(\boldsymbol{\phi}_i) = 0$ for all particles, meaning all gradients in $S_\Phi^{\mathrm{H}}$ must cancel (i.e. sum to zero). However, computing the gradient of the log-variational likelihood requires numerical estimation as analytical solutions do not exist for most models. Hence, we cannot expect the inference converges with noisy gradient estimations as the Stein force will compensate for the error in the gradient by a counterforce in the next iteration. Therefore, SMs require good (i.e. low relative variance) gradient approximations; otherwise, the particles will fluctuate around a fixed point without reaching it. We demonstrate that replacing the log hierarchical likelihood with the evidence lower bound (ELBO) can provide better (lower relative variance) gradient approximations. We call the new algorithm ELBO-within-Stein (EoS). We connect EoS with the algorithm proposed by Nalisnick & Smyth (2017) in terms of computing the gradient of different orders of the variational Rényi (VR) bound (Van Erven & Harremos, 2014). Similarly to the ELBO, the VR bound is a lower bound of the evidence, $p(\mathcal{D})$, also called the normalization constant, and is given by

$$p(\mathcal{D}) \geq \frac{1}{1-\alpha} \log \mathbb{E}_{q(\mathbf{z}|\mathcal{D},\boldsymbol{\phi})} \left[ \left( \frac{p(\mathcal{D}, \mathbf{z})}{q(\mathbf{z}|\mathcal{D}, \boldsymbol{\phi})} \right)^{1-\alpha} \right], \tag{4}$$

where $\alpha \geq 0$ is known as its order[1]. Understanding the inference of SMs in terms of the VR bound yields insight into the behavior of the two algorithms, as we can now understand $\alpha$ as controlling the variance of each component of the guide. Furthermore, presuming accurate gradient approximation for all (viable) values of $\alpha$, the connection leads to a family of inference algorithms indexed by the VR bound order.

After reviewing SVGD, the Rényi divergence, and the signal-to-noise ratio (SN-ratio) that is used to estimate the relative variance in Section 2, we make the following contributions:

- We demonstrate that inaccurate gradient estimates can lead to issues with convergence for SMs.

- We introduce a new family of inference algorithms for SMs indexed by the parameter $\alpha$. The family results from connecting inference with SMs to the Rényi $\alpha$-divergence and includes the inference algorithm by Nalisnick & Smyth (2017) as a special case for $\alpha = 0$.

- Unlike previous work, our algorithm allows for investigating a range of values for $\alpha$ for a model of interest. This allows us to investigate the convergence stability for different $\alpha$'s by measuring the SN-ratio. We find that $\alpha = 1$ is optimal for models with a latent variable

---

[1]The VR bound can be extended to $\alpha \in \mathbb{R}$. We presume $\alpha$ is finite, but we allow $\alpha$ to be less than or equal to zero (Van Erven & Harremos, 2014)

for each data point (local latent variables), resulting in better SN-ratios than all other $\alpha$ values. For models where all datapoints share a latent variable (global latent variables), using $\alpha = 0.5$ (corresponding to the Hellinger distance) is on par with Nalisnick & Smyth (2017)'s algorithm (which corresponds to $\alpha = 0$). Other values for $\alpha$ result in worse SN-ratios.

- We evaluate our inference algorithm for different values of $\alpha$ on Bayesian neural networks (BNNs) and variational autoencoders (VAEs), showing that the $\alpha$ that results in the highest performance varies depending on both model and data set.

- We describe a black-box inference algorithm for our proposed family of inference algorithms and provide a software library, called `EinSteinVI`, in NumPyro.

In Section 4 we discuss related work. We benchmark our algorithm in Section 5. Finally, we summarize our results in Section 6.

## 2 BACKGROUND

Let $\mathbf{z}$ be a latent variable of interest taking values in a space $\mathcal{Z} \subseteq \mathbb{R}^d$ (up to a diffeomorphism) and $\mathcal{D} = \{\mathbf{x}_i\}_{n \in \mathbb{N}}$ be a set of i.i.d. observations. For many models, exact Bayesian inference is computationally impracticable due to the cost of evaluating the evidence $p(\mathcal{D})$. Therefore, practitioners turn to tractable approximate variational inference (VI).

VI aims to bring a computationally cheap variational distribution $q(\mathbf{z}|\mathcal{D})$ close to the model posterior. Typically, we measure closeness by the Kullback-Leibler divergence ($D_{\mathrm{KL}}$), i.e. $D_{\mathrm{KL}}[q(\mathbf{z}|\mathcal{D}) \parallel p(\mathbf{z}|\mathcal{D})]$. However, we generally avoid directly evaluating $D_{\mathrm{KL}}[q(\mathbf{z}|\mathcal{D}) \parallel p(\mathbf{z}|\mathcal{D})]$ as this requires evaluating the evidence, $p(\mathcal{D})$. We will concern ourselves with two types of VI.

The first type of VI searches for a parameterization $\boldsymbol{\psi}^*$ of $q$ in a family of distributions $\mathcal{Q}$ that minimizes the divergence to the posterior. When the divergence is measured by $D_{\mathrm{KL}}$, this type of VI is made tractable by maximizing the evidence lower bound (ELBO), that is

$$\boldsymbol{\psi}^* = \arg\max_{\boldsymbol{\psi}} \left( \log p(\mathcal{D}) - D_{\mathrm{KL}}[q(\mathbf{z}|\mathcal{D}; \boldsymbol{\psi}) \parallel p(\mathbf{z}|\mathcal{D})] \right) = \arg\max_{\boldsymbol{\psi}} \mathbb{E}_{q(\mathbf{z}|\mathcal{D})} \left[ \log \frac{p(\mathcal{D}, \mathbf{z})}{q(\mathbf{z}|\mathcal{D}; \boldsymbol{\psi})} \right].$$

The second type of VI we consider relies on particle-based methods and is the focus of this article. This type of VI relies on transporting a finite set of particles such that their empirical measure is close to the posterior. We will discuss this method in detail below.

### 2.1 STEIN VARIATIONAL GRADIENT DESCENT

The core idea of SVGD is to perform inference by approximating the target posterior distribution $p(\mathbf{z}|\mathcal{D})$ by an empirical distribution $q_{\mathcal{Z}}(\mathbf{z}) = N^{-1} \sum_i \delta_{\mathbf{z}_i}(\mathbf{z})$ based on a set of particles $\mathcal{Z}$, where $\mathcal{Z} = \{\mathbf{z}_i\}_{i=1}^N$. One could thus see the approximating distribution $q_{\mathcal{Z}}(\mathbf{z})$ as a (uniform) mixture of point estimates, each represented by a particle $\mathbf{z}_i \in \mathcal{Z}$. The SVGD algorithm minimizes the Kullback-Leibler divergence $D_{\mathrm{KL}}[q_{\mathcal{Z}}(\mathbf{z}) \parallel p(\mathbf{z}|\mathcal{D})]$ between the approximated and the true posterior by iteratively updating the particles using the following expression:

$$\mathbf{z}_{i+1} \leftarrow \mathbf{z}_i + \epsilon S_{\mathcal{Z}}(\mathbf{z}_i)$$

where $\epsilon$ is the learning rate and $S_{\mathcal{Z}}$ denotes the Stein force.

**The two forces of SVGD** The Stein force $S_{\mathcal{Z}}$ consists of two underlying forces that work additively, with $S_{\mathcal{Z}} = S_{\mathcal{Z}}^+ + S_{\mathcal{Z}}^-$. The attractive force is given by

$$S_{\mathcal{Z}}^+(\mathbf{z}_i) = \mathbb{E}_{\mathbf{z}_j \sim q_{\mathcal{Z}}}[k(\mathbf{z}_i, \mathbf{z}_j) \nabla_{\mathbf{z}_j} \log p(\mathbf{z}_j|\mathcal{D})]$$

and the repulsive force by

$$S_{\mathcal{Z}}^-(\mathbf{z}_i) = \mathbb{E}_{\mathbf{z}_j \sim q_{\mathcal{Z}}}[\nabla_{\mathbf{z}_j} k(\mathbf{z}_i, \mathbf{z}_j)]. \tag{5}$$

Here $k : \mathbb{R}^d \times \mathbb{R}^d \to \mathbb{R}$ is a kernel. The attractive force can be seen as pushing the particles towards the modes of the true posterior distribution, smoothed by some kernel. The repulsive force stops particles with high kernel values from collapsing onto each other. In Appendix C, we demonstrate the

repulsive behavior for a radial basis function (RBF) kernel. The computational cost of $S_{\mathcal{Z}}$ is quadratic in the size of $\mathcal{Z}$, i.e. $\mathcal{O}(N^2)$, which makes SVGD computationally burdensome for high-dimensional posteriors. For a particle method such as SVGD, the number of particles required to represent a posterior distribution adequately is exponential in its dimensionality.

SVGD suffers from the curse of dimensionality (Ba et al., 2021), which results in variance collapse (i.e. variance is underestimated). Wang et al. (2018) demonstrates the problem with a simple factorized Gaussian, suggesting the (RBF) kernel introduces global statistical dependence driving the need for particles up for accurate representation. Ba et al. (2021) demonstrate that the collapse is due to the deterministic update of the attractive force. They do this by showing that re-sampling the particles at each iteration eliminates the underestimation of variance. Note that their particle re-sampling scheme by Ba et al. (2021) is not generally tractable; hence it does not suffice as a practical solution.

## 2.2 RÉNYI DIVERGENCE AND THE VARIATIONAL RÉNYI BOUND

The Rényi divergence (Rényi, 1961) is a family of divergences between distributions $p$ and $q$ indexed by the order parameter $\{\alpha | \alpha \in \mathbb{R}^+ / \{0, 1\}, |D_\alpha| < \infty\}$. The divergence is given by

$$D_\alpha \left[ p \parallel q \right] = \frac{1}{\alpha - 1} \log \int p(\mathbf{z})^\alpha q(\mathbf{z})^{1-\alpha} d\mathbf{z}.$$

The Rényi divergence can be extended to $\alpha \in \{0, 1, \infty\}$ by continuity. In addition, if we allow for $D_\alpha[p \parallel q] \leq 0$, the order can be further extended to $\alpha \in \mathbb{R}$ (Van Erven & Harremos, 2014). Several orders correspond to known divergences (see (Van Erven & Harremos, 2014) and (Li & Turner, 2016) for an overview). In particular, $\alpha = 1$ corresponds to $D_{\text{KL}}$.

Analogous to the use of the $D_{\text{KL}}$ in the ELBO, $D_\alpha$ leads to a variational Rényi bound (Li & Turner, 2016) which, when formulated as used with SMs, is given by

$$\log p(\mathcal{D}) - D_\alpha \left[ q(\mathbf{z}|\mathcal{D}, \boldsymbol{\phi}) \parallel p(\mathbf{z}|\mathcal{D}) \right] = \frac{1}{1-\alpha} \log \mathbb{E}_{q(\mathbf{z}|\boldsymbol{\phi})} \left[ \left( \frac{p(\mathbf{z}, \mathcal{D})}{q(\mathbf{z}|\mathcal{D}, \boldsymbol{\phi})} \right)^{1-\alpha} \right]. \tag{6}$$

Note that model hyper-parameters ($\boldsymbol{\phi}$) in the variational posterior, $q(\mathbf{z}|\mathcal{D}, \boldsymbol{\phi})$, are lifted to random variables when doing inference with SMs. See Appendix A for the derivation of Equation (6). Assuming reparameterization of $\mathbf{z}$ is possible, we can approximate the gradient $\Lambda(\boldsymbol{\phi})$ of Equation (6) using Monte Carlo integration by

$$\Lambda_K(\boldsymbol{\phi}) = \sum_{k=1}^{K} \boldsymbol{\omega}_k^\alpha(\mathbf{Z}, \mathcal{D}) \nabla_{\boldsymbol{\phi}} \log \left( \frac{p(\mathbf{Z}_k, \mathcal{D})}{q(\mathbf{Z}_k|\mathcal{D}, \boldsymbol{\phi})} \right), \quad \text{with } \mathbf{Z}_k \sim q(\mathbf{z}|\mathcal{D}, \boldsymbol{\phi}), \tag{7}$$

where $K \in \mathbb{N}$ number of draws used to compute the VR bound and

$$\boldsymbol{\omega}_k^\alpha(\mathbf{z}, \mathcal{D}) = \frac{1}{C} \left( \frac{p(\mathcal{D}, \mathbf{z}_k)}{q(\mathbf{z}_k|\mathcal{D}, \boldsymbol{\phi})} \right)^{1-\alpha}, \text{ with } C = \sum_{i=1}^{K} \left( \frac{p(\mathcal{D}, \mathbf{z}_i)}{q(\mathbf{z}_i|\mathcal{D}, \boldsymbol{\phi})} \right)^{1-\alpha}. \tag{8}$$

We provide the derivation in Appendix A.

## 2.3 THE SIGNAL-TO-NOISE RATIO

The signal-to-noise (SN) ratio was introduced by Rainforth et al. (2018) to study the effect of tighter variational bounds on gradient estimation. The SNR is given by

$$\text{SNR}_{M,K}(\boldsymbol{\phi}) = \left| \frac{\mathbb{E} \left[ \Delta_{M,K}^\alpha(\boldsymbol{\phi}) \right]}{\sigma \left[ \Delta_{M,K}^\alpha(\boldsymbol{\phi}) \right]} \right|, \tag{9}$$

where $\sigma[\cdot]$ is the standard deviation, $M, K \in \mathbb{N}$ are the number of Monte Carlo draws, and $\Delta_{M,K}^\alpha(\boldsymbol{\phi})$ derives from rewriting Equation (7) in the form

$$\Delta_{M,K}^\alpha(\boldsymbol{\phi}) = \frac{1}{1-\alpha} \frac{1}{M} \sum_{m}^{M} \nabla_{\boldsymbol{\phi}} \log \left[ \frac{1}{K} \sum_{k=1}^{K} \left( \frac{p(\mathbf{Z}_{m,k}, \mathcal{D})}{q(\mathbf{Z}_{m,k}|\mathcal{D}, \boldsymbol{\phi})} \right)^{1-\alpha} \right]. \tag{10}$$

Here, we separate tightening the bound (by increasing $K$) from reducing the noise in the gradient estimation (by increasing $M$). If the rate at which the expected gradient decreases is faster than the rate of decrease of the variance, the gradient estimates worsen as $K$ increases. The counter-intuitive implication is that a tighter bound can worsen the gradient estimation.

## 2.4 The Stein mixture

Variational inference with SMs (Nalisnick & Smyth, 2017) approximates the target posterior distribution $p(\mathbf{z}|\mathcal{D})$ by letting the Stein particles $\Phi = \{\boldsymbol{\phi}_i\}_{i=1}^N$ parameterize guide programs, $q(\mathbf{z}|\boldsymbol{\phi}_i, \mathcal{D})$. A SM yields a mixture marginal variational posterior, $p(\mathbf{z}|\mathcal{D}) \approx {}^{1}/{|\Phi|} \sum_{\boldsymbol{\phi} \in \Phi} q(\mathbf{z}|\boldsymbol{\phi}, \mathcal{D})$, from which it takes its name. Formally, SM is a hierarchical variational model (HVM) (Ranganath et al., 2016) with an empirical measure of particles $q_\Phi$ (defined in the same way as $q_\mathcal{Z}$) as its variational posterior, a uniform variational prior, and variational likelihood $\mathbb{E}_{q(\mathbf{z}|\mathcal{D}, \boldsymbol{\phi})} [{}^{p(\mathcal{D}, \mathbf{z}|\boldsymbol{\phi})}/{q(\mathbf{z}|\mathcal{D}, \boldsymbol{\phi})}]$. Similarly to SVGD, SM minimizes $D_{\mathrm{KL}}(q(\boldsymbol{\phi}) \| p(\boldsymbol{\phi}|\mathcal{D}))$ by iteratively transporting the particles according to the following expression

$$\boldsymbol{\phi}_{i+1} \leftarrow \boldsymbol{\phi}_i + \epsilon S_\Phi^{\mathrm{H}}(\boldsymbol{\phi}_i)$$

where $\epsilon \geq 0$ is the learning rate and $S_\Phi^{\mathrm{H}}$ is the hierarchical Stein force.

**The attractive force of SM** Like SVGD, SM also makes use of two additive forces, $S_\Phi^{\mathrm{H}} = S_\Phi^{\mathrm{H}+} + S_\Phi^-$. The repulsive force $S_\Phi^-$ is the same as in SVGD, given by Equation (5). The attractive force is given by

$$S_\Phi^{\mathrm{H}+}(\boldsymbol{\phi}_i) = \mathbb{E}_{\boldsymbol{\phi} \sim q_\Phi} \left[ k(\boldsymbol{\phi}_i, \boldsymbol{\phi}) \nabla_{\boldsymbol{\phi}} \log \mathbb{E}_{q(\mathbf{z}|\boldsymbol{\phi})} \left[ \frac{p(\mathcal{D}, \mathbf{z})}{q(\mathbf{z}|\mathcal{D}, \boldsymbol{\phi})} \right] \right],$$

where $k : \mathbb{R}^d \times \mathbb{R}^d \to \mathbb{R}$ is a kernel. From the construction of SVGD, we require that the kernel has the reproducing property, so the kernel is dense in the space of continuous functions. If we choose Gaussian guides, the expected likelihood (EL) kernel (Jebara et al., 2004) is a natural choice because it accounts for the geometry of $q(\mathbf{z}|\mathcal{D}, \boldsymbol{\phi}_j)$ and reduces to the RBF kernel for fixed variance, which is a reproducing kernel. The EL kernel is given by

$$k(\boldsymbol{\phi}_i, \boldsymbol{\phi}_j) = \int q(\mathbf{z}|\mathcal{D}, \boldsymbol{\phi}_i) q(\mathbf{z}|\mathcal{D}, \boldsymbol{\phi}_j) d\mathbf{z} = \langle q(\mathbf{z}|\mathcal{D}, \boldsymbol{\phi}_i), q(\mathbf{z}|\mathcal{D}, \boldsymbol{\phi}_j) \rangle_{L_2},$$

where $L_2$ is an inner product and $k$ is a positive definite kernel.

## 3 $\alpha$-indexed Stein mixtures inference and ELBO-within-Stein

To see the connection between the hierarchical Stein force given in Equation (3) and the Rényi divergence, consider the gradient of the log hierarchical likelihood (that occurs in $S_\Phi^{\mathrm{H}+}$) and the VR bound given in Equation (6) for $\alpha = 0$. Presuming the support of the variational likelihood $q(\mathbf{z}|\boldsymbol{\phi})$ is a subset of the support of the prior of $p$, $\mathrm{supp}(q(\mathbf{z}|\boldsymbol{\phi})) \subseteq \mathrm{supp}(p(\mathbf{z}))$, the gradient of the log hierarchical likelihood is given by

$$\begin{aligned}
\nabla_{\boldsymbol{\phi}} \log p(\mathcal{D}|\boldsymbol{\phi}) &= \nabla_{\boldsymbol{\phi}} \log \mathbb{E}_q \left[ \frac{p(\mathcal{D}, \mathbf{z})}{q(\mathbf{z}|\mathcal{D}, \boldsymbol{\phi})} \right] && (\alpha = 0, eq.\ (6)) \\
&= \nabla_{\boldsymbol{\phi}} \left( \log p(\mathcal{D}) - D_{\alpha=0}[q(\mathbf{z}|\mathcal{D}, \boldsymbol{\phi}) \| p(\mathbf{z}|\mathcal{D})] \right) \\
&= -\nabla_{\boldsymbol{\phi}} D_{\alpha=0}[q(\mathbf{z}|\mathcal{D}, \boldsymbol{\phi}) \| p(\mathbf{z}|\mathcal{D})]. && (11)
\end{aligned}$$

From Equation (11), we see that the gradient of the log marginal likelihood is exactly the gradient of the difference between the score $\log p(\mathcal{D})$, on the one hand, and the Rényi divergence (at $\alpha = 0$) between the variational posterior $q(\mathbf{z}|\boldsymbol{\phi})$ and the model posterior $p(\mathbf{z}\|\mathcal{D})$, on the other hand. Thus, Equation (11) shows that the attractive hierarchical force ($S_\Phi^{\mathrm{H}+}$) pushes the components of the variational posterior, $q(\mathbf{z}|\mathcal{D}, \boldsymbol{\phi})$, towards the model posterior, $p(\mathbf{z}|\mathcal{D})$, see Appendix D for details. The equivalence in Equation (11) suggests a whole class of hierarchical attractive forces indexed by the order $\alpha$ of the VR bound. Note that choosing $\alpha \neq 0$ means we lose the interpretation of the attractive force as moving the particles towards the nearest peak of the conditional evidence. Assuming our

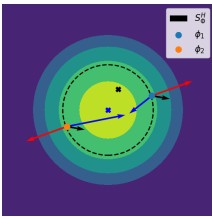

(a) Low accuracy gradient approximation

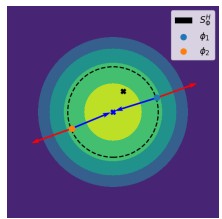

(b) High accuracy gradient approximation

Figure 1: Two particle system at theoretical fixed point. The blue arrows indicate the magnitude and direction of the attractive force, the red arrows show the repulsive force, and the black arrows the Stein force. Note that Figure 1b has no Stein force as expected for a converged system.

marginal variational posterior $q(\mathbf{z}|\boldsymbol{\phi})$ is reparameterizable, we can approximate the attractive force for any $\alpha \geq 0$ as

$$S_\Phi^{\alpha+}(\boldsymbol{\phi}_i) = \mathbb{E}_{\boldsymbol{\phi} \sim q_\Phi}\left[k(\boldsymbol{\phi}_i, \boldsymbol{\phi})\Lambda_K(\boldsymbol{\phi})\right], \tag{12}$$

where $\Lambda_K(\phi)$ is given by Equation (7). We call inference with Equation (12) $\alpha$-indexed Stein mixture inference. There are two special cases of $\alpha$ that are worth highlighting. The first is $\alpha = 1/2$, for which the Rényi divergence corresponds to the Hellinger divergence (Van Erven & Harremos, 2014)(Li & Turner, 2016). The second is $\alpha = 1$, corresponding to the $D_{\text{KL}}$-divergence. In this case, the VR bound recovers the ELBO. We call this instance of our $\alpha$-indexed SM inference algorithm ELBO-within-Stein. In Appendix E we show that we can also recover the $\alpha = 1$ case directly by applying Jensen's inequality to the conditional evidence.

## 3.1 INVESTIGATING THE SIGNAL-TO-NOISE RATIO

Estimation of a Stein mixture converges when $S_\Phi^H = 0$, which means that the repulsive and attractive forces must be equal and oppose for them to cancel. Hence, convergence requires $\Delta_{M,K}^\alpha(\boldsymbol{\phi})$ and $\nabla_{\boldsymbol{\phi}_1} k(\boldsymbol{\phi}_1, \boldsymbol{\phi}_2)$ to be accurate. In Figure 1 we demonstrate the effect of inaccurate gradient approximations. To study the sensitivity of gradient approximations to the choice of $\alpha$, we measure the SN-ratio (see Equation (9)) of the VR bound gradients (see Equation (6)). We simulate data $\{\mathbf{x}_i\}_{i=1}^n$ from a simple latent variable model given by $\mathcal{N}(\mathcal{D}|\mathbf{z}, I_d)\mathcal{N}(\mathbf{z}|\boldsymbol{\mu}, I_d)$, where $\boldsymbol{\mu} \in \mathbb{R}^d$ is unknown and $I_d$ is the $d$-dimensional identity matrix. To approximate its posterior we use a Stein mixture of the form

$$1/2(\mathcal{N}(\boldsymbol{\phi}_1, 3/2I_d) + \mathcal{N}(\boldsymbol{\phi}_2, 3/2I_d)),$$

and an expected likelihood kernel. We choose a (computationally convenient) fixed variance such that the Stein mixture cannot exactly recover the posterior. We can see this as the posterior is unimodal which is only the case for the Stein mixture if $|\boldsymbol{\phi}_1 - \boldsymbol{\phi}_2| < 3$ (Behboodian, 1970), but in this interval the variance of the Stein mixture will be greater than or equal to $3/2$. With the expected likelihood kernel we can analytically characterize all fixed points for the Stein particles as

$$-\nabla_{\boldsymbol{\phi}_1} \frac{1}{1-\alpha} \log \mathbb{E}_{q(\mathbf{z}|\boldsymbol{\phi}_1)}\left[\left(\frac{p(\mathbf{z}, \mathcal{D})}{q(\mathbf{z}|\boldsymbol{\phi}_1)}\right)^{1-\alpha}\right] = \nabla_{\boldsymbol{\phi}_2} \frac{1}{1-\alpha} \log \mathbb{E}_{q(\mathbf{z}|\boldsymbol{\phi}_2)}\left[\left(\frac{p(\mathbf{z}, \mathcal{D})}{q(\mathbf{z}|\boldsymbol{\phi}_2)}\right)^{1-\alpha}\right].$$

See the Appendix B for the derivation. To measure the effect of gradient approximation on the system we use Equation (10) to estimate the gradients.

To conduct our experiment, we sample the location $\boldsymbol{\mu}$ from a 20-dimensional standard Gaussian and use this $\boldsymbol{\mu}$ to simulate $n = 64$ data points $\mathcal{D}$. We then approximate the gradients at a random point close to a fixed point

$$(\boldsymbol{\phi}_1, \boldsymbol{\phi}_2) = \left(\frac{\boldsymbol{\mu} + n\overline{\mathcal{D}}}{n+1} + \nabla_{\boldsymbol{\phi}_1}\Delta_{M,K}^\alpha(\boldsymbol{\phi}_1) + \epsilon, \frac{\mu + n\overline{\mathcal{D}}}{n+1} + \nabla_{\boldsymbol{\phi}_2}\Delta_{M,K}^\alpha(\boldsymbol{\phi}_2)\right),$$

where $\overline{\mathcal{D}}$ is the data average, $\mu+n\overline{\mathcal{D}}/n+1$ is the posterior mean and $\epsilon$ offsets each dimension by a Gaussian with mean zero and variance 0.01.

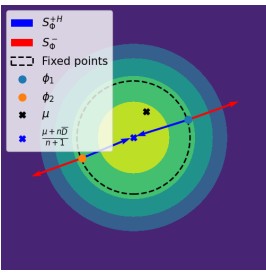

(a) Experimental setup.

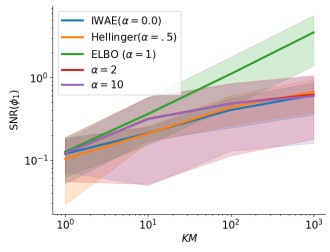

(b) SN-ratio convergence with a local latent variable.

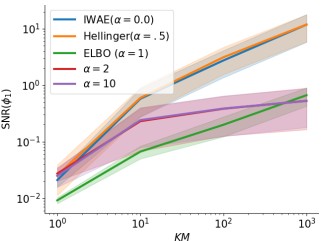

(c) SN-ratio convergence with a global latent variable.

Figure 2b and Figure 2c show the convergence of the SN-ratio (see Equation (9)) as we tighten the VR bound by increasing either $K$ or $M$. We only show the SN-ratio for the first particle ($\boldsymbol{\phi}_1$) as the behaviour for the second particle is the same. For the ELBO ($\alpha = 1$), we fix $K = 1$ and increase $M$ to reduce the gradient approximation variance, while for the rest ($\alpha \in \{0, 0.5, 2, 10\}$), we fix $M = 1$ and increase $K$ to tighten the VR bound. In Figure 2b, there is a latent variable for each data point. Note how the SN-ratio only improves with tightness for $\alpha = 1$ (green line). In Figure 2c all data points share a latent variable.

Figure 2a illustrates the experimental setup in two dimensions. For legibility in Figure 2a, we do not include the perturbation (i.e. added $\epsilon$ noise) on $\boldsymbol{\phi}_1$ to the visualization. The contours correspond to the exact posterior. As the particles are placed equidistant from the posterior mean (marked with a blue cross), in this setting the Stein forces are zero. As we would expect (see blue arrows in Figure 2a) the gradient estimations of $\nabla_{\boldsymbol{\phi}} \Delta_{M,K}^{\alpha}$ are equal and opposite for the two particles.

For $\alpha \neq 1$ we fix $M = 1$ and vary $K$, while for $\alpha = 1$ we fix $K = 1$ and vary $M$. We do not need to consider $\alpha \neq 1$ when $K = 1$ as the associated gradient scaling $(1 - \alpha)$ cancels in the SN-ratio. We empirically estimate the SN-ratio by estimating the expectation and standard deviation from $10,000$ gradient samples. In Figure 2b we show a local variate of the model where there is a latent variable $\mathbf{z}_i$ for each $\mathbf{x}_i \in \mathcal{D}$. We see that for $\alpha \neq 1$ the SN-ratio does not depend on the particular choice of $\alpha$ and that the growth in SN-ratio is superseded by that of $\alpha = 1$. This means there is little to no benefit in increasing $K$ beyond $K = 1$, for which we recover the ELBO gradient when the guide is reparameterizable (see Section 2). In Figure 2c we evaluate a global latent variable variate of the model, so that there is one $\mathbf{z}$ for all datapoints in $\mathcal{D}$. As with the local version, we fix either $M$ or $K$. For this model, we see that $\alpha = 0$ and $\alpha = 1/2$ achieve the highest SN-ratio. The result aligns with the BNN example, where $\alpha = 1$ is not dominating in performance over the $\alpha \in \{0, 1/2\}$ on all datasets.

High precision is desirable to avoid fluctuation at convergence. From the above results, we see that $\alpha = 0$ is not necessarily the best choice for precise (high SN-ratio) gradient estimation. In particular, for local latent variable models, $\alpha = 1$ is a better choice, and for that, for global latent models, $\alpha = 1/2$ is on par with $\alpha = 0$ in our experiment.

### 3.2 BLACK-BOX INFERENCE FOR ELBO-WITHIN-STEIN

We provide a mini-batch version of ELBO-within-Stein, called EinSteinVI, in NumPyro. To compute the VR bound exactly requires all the datapoints, that is, we cannot represent the bound as a point-wise expectation, except for $\alpha = 1$. Therefore, in order to make EinSteinVI scalable to tall data, we provide an approximation of the VR bound which replaces the likelihood by $p_{\mathcal{I}}(\mathcal{D}|\mathbf{z}, \boldsymbol{\phi}) = \prod_{i \in \mathcal{I}} p(x_i|\mathbf{z}, \boldsymbol{\phi})^{|\mathcal{D}|/|\mathcal{I}|}$, where $\mathcal{I}$ is a subset of a permutation of the data indices. The approximate attractive force $S_{\Phi}^{\alpha+}(\boldsymbol{\phi})$ used in EinSteinVI is given by

$$S_{\Phi}^{\alpha+}(\boldsymbol{\phi}_i) = \mathbb{E}_{\boldsymbol{\phi} \sim q_{\Phi}} \left[ k(\boldsymbol{\phi}, \boldsymbol{\phi}_i) \nabla_{\boldsymbol{\phi}} \frac{1}{1 - \alpha} \log \mathbb{E}_{q_{\mathcal{I}}(\mathbf{z}|\mathcal{D})} \left[ \left( \frac{p_{\mathcal{I}}(\mathcal{D}|\mathbf{z}, \boldsymbol{\phi}) p(\mathbf{z})}{q_{\mathcal{I}}(\mathbf{z}|\mathcal{D}, \boldsymbol{\phi})} \right)^{1-\alpha} \right] \right], \quad (13)$$

which recovers the exact VR bound when $|\mathcal{I}| = |\mathcal{D}|$. We describe the NumPyro integration and provide example programs in Appendix I.

Table 1: Average RMSE (lower is better) test results for BNNs on UCI benchmark. EoS (ours) corresponds $\alpha = 1$, Hell (ours) to $\alpha = 0.5$, and SM (Nalisnick & Smyth, 2017) to $\alpha = 0$. Parentheses mean we use Dirac delta guides. EoS and Hell gives the best results.

| Dataset | Average Root Mean Squared Deviation (RMSE) | | | | | |
| | EoS | Hell | SM | SVGD | MFVI | Laplace |
|---|---|---|---|---|---|---|
| Boston | $3.5 \pm 0.82$ $(2.8 \pm 0.4)$ | $3.92 \pm 1.3$ $(\mathbf{2.71 \pm 0.26})$ | $3.9 \pm 1.29$ $(2.715 \pm 0.263)$ | $2.86 \pm 0.23$ | $3.28 \pm 0.1$ | $3.68 \pm 0.33$ |
| Concrete | $5.76 \pm 0.55$ $(\mathbf{4.61 \pm 0.34})$ | $6.55 \pm 0.63$ $(5.2 \pm 0.3)$ | $6.51 \pm 0.59$ $(5.23 \pm 0.32)$ | $5.54 \pm 0.33$ | $5.6 \pm 0.3$ | $5.22 \pm 0.43$ |
| Energy | $0.53 \pm 0.05$ $(\mathbf{0.45 \pm 0.03})$ | $0.94 \pm 0.15$ $(0.67 \pm 0.03)$ | $0.81 \pm 0.16$ $(0.74 \pm 0.05)$ | $1.30 \pm 0.08$ | $1.75 \pm 0.15$ | $0.46 \pm 0.03$ |
| Naval | $0.04 \pm 0.04$ $(\mathbf{0.00 \pm 0.00})$ | $0.004 \pm 0.001$ $(0.001 \pm 0.00)$ | $0.004 \pm 0.002$ $(0.001 \pm 0.000)$ | $0.007 \pm 0.000$ | $\mathbf{0.000 \pm 0.00}$ | $\mathbf{0.00 \pm 0.00}$ |
| Wine | $0.6 \pm 0.038$ $(\mathbf{0.07 \pm 0.00})$ | $0.61 \pm 0.03$ $(0.08 \pm 0.00)$ | $0.61 \pm 0.03$ $0.08 \pm 0.00$ | $0.62 \pm 0.04$ | $0.59 \pm 0.04$ | $0.61 \pm 0.01$ |
| Yacht | $1.76 \pm 0.41$ $(\mathbf{0.45 \pm 0.03})$ | $1.66 \pm 0.65$ $(0.67 \pm 0.03)$ | $1.61 \pm 0.5$ $(0.74 \pm 0.05)$ | $1.11 \pm 0.3$ | $4.09 \pm 0.34$ | $2.16 \pm 0.37$ |
| Power | $4.04 \pm 0.16$ $(\mathbf{3.91 \pm 0.18})$ | $4.15 \pm 0.21$ $(3.98 \pm 0.19)$ | $4.16 \pm 0.21$ $(3.97 \pm 0.2)$ | $4.06 \pm 0.17$ | $3.94 \pm 0.18$ | $3.99 \pm 0.17$ |

## 4 RELATED WORK

Nalisnick & Smyth (2017) first suggested Stein mixtures as an alternative to HVMs (Ranganath et al., 2016). Using SVGD allows Stein mixtures to side-step the need of HVMs for an auxiliary distribution to keep the bound (learning objective) tight. This is an improvement as the effect of the auxiliary distribution on the approximation is implicit and therefore hard to understand, whereas with Stein mixtures the choice of kernel controls the tightness and we have theoretical understanding of kernels (Wang et al., 2019; Gorham & Mackey, 2017; Liu & Wang, 2018). Mixture approximations have a long history of work (Jaakkola & Jordan, 1998; Bishop et al., 1997; Gershman et al., 2012; Miller et al., 2017) focusing on approximating or lower-bounding the intractable mixture ELBO. Van Erven & Harremos (2014) unifies a number of variational techniques by considering them as optimizing different orders of the VR bound. They further demonstrate that two different variants of mini-batch training with the VR bound recover Stochastic EP (Li et al., 2015) and Black-box $\alpha$ (Hernandez-Lobato et al., 2016), respectively. The Rényi divergence has been studied in other forms under the name $\alpha$-divergence (Amari, 2012; Tsallis, 1988). Hernandez-Lobato et al. (2016) introduced a black-box algorithm for variational inference based on the $\alpha$-divergence using automatic differentiation. Unlike our algorithm, their algorithm is not for HVMs. Rainforth et al. (2018) demonstrated that for VAEs the gradient estimation degrades for multi-sample approximations when using the importance weighted variational autoencoder (IWAE) bound (Burda et al., 2015). Furthermore, Rainforth et al. (2018) showed that this is not the case when using the ELBO. Rainforth et al. (2018) differs from our work in that the VAEs estimated are with a point mass guide, as their inference algorithm is not for HVMs. Le et al. (2020) investigates the deterioration experimentally, providing evidence for it on several real world tasks. Tucker et al. (2018) show that by double reparameterizing the gradient estimator, they can eliminate the degrading SN-ratio for multi-sample estimation of the IWAE gradient, among others.

## 5 EXAMPLES

We evaluate $\alpha$-indexed Stein mixture inference by inferring Bayesian neural networks (BNN) and variational autoencoders (VAE) on standard datasets. We use the BNNs for regression on the UCI regression benchmark (the same as Hernández-Lobato & Adams (2015)) and VAE for unsupervised learning on MNIST (Salakhutdinov & Murray, 2008; LeCun et al., 1998) and OMNIGLOT (Lake et al., 2013).

**Bayesian neural networks** For brevity, we present BNNs for the subset of the UCI regression benchmark detailed in Appendix G. All datasets use real-valued features. We use a 90-10 split for training and test datasets. We compare $\alpha$-indexed SM inference for $\alpha \in \{0, 0.5, 1\}$ on BNN regression. We test with two guides: factorized Gaussian guides with an EL kernel and point mass (Dirac delta) guides with an RBF kernel. Like Liu & Wang (2016), we use a BNN with one hidden layer of size fifty and a `RELU` activation. We put a Gamma$(1, 0.1)$ prior on the precision of the neurons and the likelihood. We use five particles for all experiments. We run all datasets for 35,000

Table 2: Average log likelihood (higher is better) test results for BNNs on UCI benchmark. EoS (ours) corresponds $\alpha = 1$, Hell (ours) to $\alpha = 0.5$, and SM (Nalisnick & Smyth, 2017) to $\alpha = 0$. Parentheses mean we use Dirac delta guides. EoS generally outperforms Hell and SM.

| Dataset | Average log-likelihood | | | | | |
| | EoS | Hell | SM | SVGD | MFVI | Laplace |
| --- | --- | --- | --- | --- | --- | --- |
| Boston | $-0.66 \pm 0.34$ $(-0.64 \pm 0.27)$ | $-0.79 \pm 0.32$ $(-1.50 \pm 1.34)$ | $-0.79 \pm 0.33$ $(-1.48 \pm 1.32)$ | $-2.55 \pm 0.08$ | $-0.74 \pm 0.04$ | $\mathbf{-0.56 \pm 0.03}$ |
| Concrete | $-0.58 \pm 0.25$ $(\mathbf{-0.5 \pm 0.09})$ | $-1.02 \pm 0.37$ $(-1 \pm 0.27)$ | $-1.01 \pm 0.37$ $(-1 \pm 0.29)$ | $-3.18 \pm 0.06$ | $-0.54 \pm 0.11$ | $-0.96 \pm 0.49$ |
| Energy | $0.02 \pm 0.76$ $(-0.25 \pm 0.81)$ | $-0.10 \pm 0.63$ $(-3.02 \pm 1.64)$ | $-0.13 \pm 0.70$ $(-1.61 \pm 1.50)$ | $-1.76 \pm 0.03$ | $0.06 \pm 0.12$ | $\mathbf{0.31 \pm 0.70}$ |
| Naval | $1.72 \pm 0.84$ $(-21.40 \pm 19.71)$ | $-0.54 \pm 0.49$ $(-5.32 \pm 0.96)$ | $-0.99 \pm 0.78$ $(-4.80 \pm 1.04)$ | $\mathbf{3.46 \pm 0.04}$ | $1.48 \pm 1.37$ | $2.05 \pm 0.93$ |
| Wine | $-1.38 \pm 0.01$ $(-1.23 \pm 0.05)$ | $-1.35 \pm 0.04$ $(-1.43 \pm 0.11)$ | $-1.35 \pm 0.04$ $(-1.43 \pm 0.10)$ | $\mathbf{-0.96 \pm 0.5}$ | $-1.26 \pm 0.07$ | $-1.52 \pm 0.08$ |
| Yacht | $\mathbf{0.08 \pm 0.46}$ $(-0.20 \pm 0.58)$ | $-0.26 \pm 0.53$ $(-2.83 \pm 2.19)$ | $-0.27 \pm 0.51$ $(-2.89 \pm 2.17)$ | $-2.11 \pm 0.63$ | $-0.83 \pm 0.05$ | $-0.12 \pm 0.10$ |
| Power | $\mathbf{0.04 \pm 0.06}$ $(-0.20 \pm 0.58)$ | $-0.08 \pm 0.10$ $(-0.23 \pm 0.18)$ | $-0.08 \pm 0.11$ $(-0.23 \pm 0.15)$ | $-2.83 \pm 0.03$ | $0.03 \pm 0.06$ | $0.02 \pm 0.07$ |

Table 3: Log likelihood (higher is better) test results for VAE. EoS (ours) corresponds $\alpha = 1$, Hell (ours) to $\alpha = 0.5$, and SM (Nalisnick & Smyth, 2017) to $\alpha = 0$.

| Dataset | SM | Hell | EoS |
| --- | --- | --- | --- |
| MNIST | $-101.874$ | $-100.541$ | $-77.400$ |
| OMNIGLOT | $-146.241$ | $-146.257$ | $-148.428$ |

epochs with a subsample size of 32, the Adam optimizer (Kingma & Ba, 2014) and a step size of 0.002. All measurements are repeated three times and obtained on a GPU[2].

We compare against the SVGD implantation from Liu et al. (2016) with 20 particles, mean field variational Bayes (MFVI) with a factorized Gaussian guide (Hoffman et al., 2013) and Laplace approximation. For the latter two we inference engines from NumPyro (Phan et al., 2019).

Table 1 shows the root mean squared error (RMSE) on test sets. EoS with delta Guides out performance baselines and other $\alpha$-orders, except on Boston. SM and Hell perform similarly with factorized Gaussian guides, which aligns with our SN-ratio experiment that shows the gradient approximations are similar for these two cases. Note the Stein mixtures use only five particles, whereas SVGD uses twenty. Table 2 gives the log-likelihood on the same test sets. EoS achieves better average log-likelihood for all datasets with factorized Gaussian guides than other $\alpha$-orders. We see that $\alpha = 0$ and $\alpha = .5$ performs similarly with a factorized Gaussian prior, which aligns with our SN-ratio experiment in that the quality of gradient approximations is similar.

**Variational autoencoder** We evaluate Stein mixtures and SVGD for VAEs on two datasets with $\alpha \in \{0, 0.5, 1\}$. We use binarized MNIST (Salakhutdinov & Murray, 2008; LeCun et al., 1998), a dataset of $28 \times 28$ pixel images of handwritten single digit numbers, and a variate of OMNIGLOT (Lake et al., 2013), which contains $28 \times 28$ pixel images of characters from fifty different alphabets. We use the same VAE architecture as Burda et al. (2015), detailed in Appendix H. For both datasets we optimize using the Adam optimizer and learning rate of $5 \cdot 10^{-4}$. We optimize with a batch size of 20 and use 20 draws to approximate the gradients. For OMNIGLOT we use 20 epochs and for MNIST we use 50. Table 3 show the performance of ELBO-within-Stein for $\alpha \in \{0.0, 0.5, 1.0\}$. We find that ELBO-within-Stein with $\alpha = 1$ achieves better log-likelihoods on MNIST datasets. On OMNIGLOT $\alpha = 0.5$ and $\alpha = 0$ achieve comparable log-likelihood with $\alpha 0$ the slightly outperforming $\alpha = 0.5$.

## 6 SUMMARY

We introduce a new algorithm called ELBO-within-Stein (EoS) based on a new connection between the inference of Stein mixtures and the Rényi variational bound. We demonstrate that Eos provides better gradient approximations than alternative algorithms, which results in better performance for standard benchmark problems. EoS is integrated as a black box library in the NumPyro PPL which is distributed freely.

---

[2]Quadro RTX 6000 with Cuda V11.4.120

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

## A  VARIATIONAL RÉNYI BOUND

For convenience, we derive the variational Rényi bound (Li & Turner, 2016) in the context of inference with our algorithm below. Recall that Stein mixtures lift the set of guide hyper-parameters $\phi$ (optimized in VI) to a random variable $\phi$. Let $\mathcal{D}$ be a finite set of observations, $\mathbf{z} \in \mathbb{R}^d$ be a latent variable, $D_\alpha [q||p]$ the Rényi $\alpha$-divergence (Rényi, 1961) between distributions $p$ and $q$, and $\phi \in \mathbb{R}^d$ a set of guide hyper-parameters lifted to a random variable. Then we have

$$\log p(\mathcal{D}) - D_\alpha [q(\mathbf{z}|\mathcal{D}, \phi)\|p(\mathbf{z}|\mathcal{D})] = \frac{1}{1-\alpha} \log \mathbb{E}_{q(\mathbf{z}|\phi)} \left[ \left( \frac{p(\mathbf{z}, \mathcal{D})}{q(\mathbf{z}|\mathcal{D}, \phi)} \right)^{1-\alpha} \right]. \qquad (14)$$

To see this, consider,

$$\begin{aligned}
D_\alpha [q(\mathbf{z}|\mathcal{D}, \phi)\|p(\mathbf{z}|\mathcal{D})] &= \frac{1}{\alpha - 1} \log \int q(\mathbf{z}|\mathcal{D}, \phi)^\alpha p(\mathbf{z}|\mathcal{D})^{1-\alpha} d\mathbf{z} \\
&= \frac{1}{\alpha - 1} \log \int q(\mathbf{z}|\mathcal{D}, \phi)^\alpha \left( \frac{p(\mathbf{z}, \mathcal{D})}{p(\mathcal{D})} \right)^{1-\alpha} d\mathbf{z} \\
&= \frac{1}{\alpha - 1} \log \int q(\mathbf{z}|\mathcal{D}, \phi)^\alpha p(\mathbf{z}, \mathcal{D})^{1-\alpha} d\mathbf{z} \cdot p(\mathcal{D})^{\alpha - 1} \\
&= \frac{1}{\alpha - 1} \log \int q(\mathbf{z}|\mathcal{D}, \phi)^\alpha p(\mathbf{z}, \mathcal{D})^{1-\alpha} d\mathbf{z} + \frac{\alpha - 1}{\alpha - 1} \log p(\mathcal{D}) \\
&= \frac{1}{\alpha - 1} \log \int q(\mathbf{z}|\mathcal{D}, \phi) q(\mathbf{z}|\mathcal{D}, \phi)^{-(1-\alpha)} p(\mathbf{z}, \mathcal{D})^{1-\alpha} + \log p(\mathcal{D}) \\
&= \frac{1}{\alpha - 1} \log \mathbb{E}_{q(\mathbf{z}|\mathcal{D}, \phi)} \left[ \left( \frac{p(\mathbf{z}, \mathcal{D})}{q(\mathbf{z}|\mathcal{D}, \phi)} \right)^{1-\alpha} \right] + \log p(\mathcal{D}),
\end{aligned}$$

from which we recover the desired equality by rearranging and multiplying both sides by negative one,

$$\log p(\mathcal{D}) - D_\alpha(q(\mathbf{z}|\boldsymbol{\phi})\|p(\mathbf{z}|\mathcal{D})) = \frac{1}{1-\alpha} \log \mathbb{E}_{q(\mathbf{z}|\mathcal{D},\boldsymbol{\phi})} \left[ \left( \frac{p(\mathbf{z},\mathcal{D})}{q(\mathbf{z}|\boldsymbol{\phi})} \right)^{1-\alpha} \right].$$

To shorten notation, we let $C^\alpha(\mathcal{D},\boldsymbol{\phi}) = \mathbb{E}\left[ \left( p(\mathbf{z},\mathcal{D})/q(\mathbf{z}|\mathcal{D},\boldsymbol{\phi}) \right)^{1-\alpha} \right]$ where the expectation is with respect to $q(\mathbf{z}|\mathcal{D},\boldsymbol{\phi})$. The gradient of the variational Rényi bound with respect to $\boldsymbol{\phi}$ is

$$\nabla_{\boldsymbol{\phi}} \frac{1}{1-\alpha} \log \mathbb{E}\left[ \left( \frac{p(\mathbf{z},\mathcal{D})}{q(\mathbf{z}|\mathcal{D},\boldsymbol{\phi})} \right)^{1-\alpha} \right] = \frac{1}{1-\alpha} C^\alpha(\mathcal{D},\boldsymbol{\phi})^{-1} \mathbb{E}\left[ \nabla_{\boldsymbol{\phi}} \left( \frac{p(\mathbf{z},\mathcal{D})}{q(\mathbf{z}|\mathcal{D},\boldsymbol{\phi})} \right)^{1-\alpha} \right]$$

$$= C^\alpha(\mathcal{D},\boldsymbol{\phi})^{-1} \mathbb{E}\left[ \left( \frac{p(\mathbf{z},\mathcal{D})}{q(\mathbf{z}|\mathcal{D},\boldsymbol{\phi})} \right)^{1-\alpha} \nabla_{\boldsymbol{\phi}} \log \left( \frac{p(\mathbf{z},\mathcal{D})}{q(\mathbf{z}|\mathcal{D},\boldsymbol{\phi})} \right) \right]$$

$$= \mathbb{E}\left[ \frac{\left( \frac{p(\mathbf{z},\mathcal{D})}{q(\mathbf{z}|\mathcal{D},\boldsymbol{\phi})} \right)^{1-\alpha}}{C^\alpha(\mathcal{D},\boldsymbol{\phi})^{-1}} \nabla_{\boldsymbol{\phi}} \log \left( \frac{p(\mathbf{z},\mathcal{D})}{q(\mathbf{z}|\mathcal{D},\boldsymbol{\phi})} \right) \right]$$

$$= \mathbb{E}\left[ \omega^\alpha(\mathbf{z},\mathcal{D}) \nabla_{\boldsymbol{\phi}} \log \left( \frac{p(\mathbf{z},\mathcal{D})}{q(\mathbf{z}|\mathcal{D},\boldsymbol{\phi})} \right) \right],$$

where

$$\omega^\alpha(\mathbf{z},\mathcal{D}) = \frac{\left( p(\mathbf{z},\mathcal{D})/q(\mathbf{z}|\mathcal{D},\boldsymbol{\phi}) \right)^{1-\alpha}}{\mathbb{E}\left[ \left( p(\mathbf{z},\mathcal{D})/q(\mathbf{z}|\mathcal{D},\boldsymbol{\phi}) \right)^{1-\alpha} \right]}.$$

## B  CHARACTERIZING TWO PARTICLE FIXED POINTS

We give the full derivation of stationary points for the Stein mixture that we consider in Section 3. Recall that Section 3 investigated the SN-ratio for a Stein mixture given by

$$\frac{1}{2} \left( \mathcal{N}(\boldsymbol{\phi}_1, 3/2 I_d) + \mathcal{N}(\boldsymbol{\phi}_2, 3/2 I_d) \right),$$

where $\boldsymbol{\phi}_1, \boldsymbol{\phi}_2 \in \mathbb{R}^d$ are two $d$-dimensional particles. We use the kernel given by

$$k(\boldsymbol{\phi}_1, \boldsymbol{\phi}_2) = \exp\left( -\frac{1}{h} \|\boldsymbol{\phi}_1 - \boldsymbol{\phi}_2\|_2^2 \right), \tag{15}$$

where $h \in \mathbb{R}^+$ is the bandwidth. The kernel has the following properties:

$$\nabla_{\boldsymbol{\phi}_1} k(\boldsymbol{\phi}_1, \boldsymbol{\phi}_2) = -\nabla_{\boldsymbol{\phi}_2} k(\boldsymbol{\phi}_1, \boldsymbol{\phi}_2),$$
$$k(\cdot, \cdot) = 1,$$
$$k(\boldsymbol{\phi}_1, \boldsymbol{\phi}_2) = k(\boldsymbol{\phi}_2, \boldsymbol{\phi}_1),$$
$$\nabla_{\boldsymbol{\phi}} k(\cdot, \cdot) = 0,$$

which we will use in the derivation. Finally, we introduce $\xi_\alpha(\boldsymbol{\phi}) = \frac{1}{1-\alpha} \log \mathbb{E}_{q(\mathbf{z}|\boldsymbol{\phi})} \left[ \left( \frac{p(\mathbf{z},\mathcal{D})}{q(\mathbf{z}|\boldsymbol{\phi})} \right)^{1-\alpha} \right]$ as notation short-hand.

Our two particle configuration reaches a fixed point when

$$(\boldsymbol{\phi}_1 + \epsilon S_\Phi^H(\boldsymbol{\phi}_1), \boldsymbol{\phi}_2 + \epsilon S_\Phi^H(\boldsymbol{\phi}_2)) = (\boldsymbol{\phi}_1, \boldsymbol{\phi}_2),$$

where $\epsilon \geq 0$ is the step size. Therefore, $S_\Phi^H(\boldsymbol{\phi}_1) = 0$ and $S_\Phi^H(\boldsymbol{\phi}_2) = 0$ at any fixed point. $S_\Phi^H(\boldsymbol{\phi}_1)$ is given by

$$S_\Phi^H(\boldsymbol{\phi}_1) = \overbrace{k(\boldsymbol{\phi}_1, \boldsymbol{\phi}_1)}^{1} \nabla_{\boldsymbol{\phi}_1} \xi_\alpha(\boldsymbol{\phi}_1) + \overbrace{\nabla_{\boldsymbol{\phi}_1} k(\boldsymbol{\phi}_1, \boldsymbol{\phi}_1)}^{0} + k(\boldsymbol{\phi}_1, \boldsymbol{\phi}_2) \nabla_{\boldsymbol{\phi}_2} \xi_\alpha(\boldsymbol{\phi}_2) + \nabla_{\boldsymbol{\phi}_2} k(\boldsymbol{\phi}_1, \boldsymbol{\phi}_2)$$

$$= \nabla_{\boldsymbol{\phi}_1} \xi_\alpha(\boldsymbol{\phi}_1) + k(\boldsymbol{\phi}_1, \boldsymbol{\phi}_2) \nabla_{\boldsymbol{\phi}_2} \xi_\alpha(\boldsymbol{\phi}_2) + \nabla_{\boldsymbol{\phi}_2} k(\boldsymbol{\phi}_1, \boldsymbol{\phi}_2) = 0.$$

Therefore,

$$-\nabla_{\boldsymbol{\phi}_2}k(\boldsymbol{\phi}_1,\boldsymbol{\phi}_2)=\nabla_{\boldsymbol{\phi}_1}\xi_\alpha(\boldsymbol{\phi}_1)+k(\boldsymbol{\phi}_1,\boldsymbol{\phi}_2)\nabla_{\boldsymbol{\phi}_2}\xi_\alpha(\boldsymbol{\phi}_2) \tag{16}$$

at a fixed point. By a similar argument for $S_\Phi^H(\boldsymbol{\phi}_2)$, we have

$$\nabla_{\boldsymbol{\phi}_1}k(\boldsymbol{\phi}_1,\boldsymbol{\phi}_2)=-(\nabla_{\boldsymbol{\phi}_2}\xi_\alpha(\boldsymbol{\phi}_2)+k(\boldsymbol{\phi}_1,\boldsymbol{\phi}_2)\nabla_{\boldsymbol{\phi}_1}\xi_\alpha(\boldsymbol{\phi}_2)) \tag{17}$$

at a fixed point. As $\nabla_{\boldsymbol{\phi}_1}k(\boldsymbol{\phi}_1,\boldsymbol{\phi}_2)=-\nabla_{\boldsymbol{\phi}_2}k(\boldsymbol{\phi}_1,\boldsymbol{\phi}_2)$, it follows from Equations (16) and (17) that

$$\nabla_{\boldsymbol{\phi}_1}k(\boldsymbol{\phi}_1,\boldsymbol{\phi}_2)=-\nabla_{\boldsymbol{\phi}_2}k(\boldsymbol{\phi}_1,\boldsymbol{\phi}_2)$$
$$-(\nabla_{\boldsymbol{\phi}_2}\xi_\alpha(\boldsymbol{\phi}_2)+k(\boldsymbol{\phi}_1,\boldsymbol{\phi}_2)\nabla_{\boldsymbol{\phi}_1}\xi_\alpha(\boldsymbol{\phi}_2))=\nabla_{\boldsymbol{\phi}_1}\xi_\alpha(\boldsymbol{\phi}_1)+k(\boldsymbol{\phi}_1,\boldsymbol{\phi}_2)\nabla_{\boldsymbol{\phi}_2}\xi_\alpha(\boldsymbol{\phi}_2)$$
$$-\nabla_{\boldsymbol{\phi}_2}\xi_\alpha(\boldsymbol{\phi}_2)(1+k(\boldsymbol{\phi}_1,\boldsymbol{\phi}_2))=\nabla_{\boldsymbol{\phi}_1}\xi_\alpha(\boldsymbol{\phi}_1)(1+k(\boldsymbol{\phi}_1,\boldsymbol{\phi}_2))$$
$$-\nabla_{\boldsymbol{\phi}_2}\xi_\alpha(\boldsymbol{\phi}_2)=\nabla_{\boldsymbol{\phi}_1}\xi_\alpha(\boldsymbol{\phi}_1).$$

Hence, we see that at any fixed point for our two particle configuration, the gradients of the VR-bound are equal and opposite.

## C    KERNELS IN SVGD

For an example of a kernel, consider the radial basis function (RBF) kernel $k(\mathbf{z}_i,\mathbf{z}_j)=\exp\left(-\frac{1}{h}\parallel\mathbf{z}_i-\mathbf{z}_j\parallel_2^2\right)$ with bandwidth parameter $h$, chosen as $\frac{1}{\log N}\mathrm{med}(\mathcal{Z})$, where med is the median operator.

The repulsive force moves particles away from each other, ensuring that they do not collapse onto the same mode. For the RBF kernel, the repulsive force becomes

$$\mathbb{E}_{\mathbf{z}_j\sim q_{\mathcal{Z}}}\left[\nabla_{\mathbf{z}_j}k(\mathbf{z}_i,\mathbf{z}_j)\right]=\sum_j-\frac{2}{h}k(\mathbf{z}_i,\mathbf{z}_j)\left(\mathbf{z}_i-\mathbf{z}_j\right)\cdot\mathbf{1_d},$$

where $\cdot$ is the (euclidean) inner product and $\mathbf{1_d}$ is a d-dimensional one vector. It follows that $\mathbf{z}_i$ is pushed away from $\mathbf{z}_j$ when $k(\mathbf{z}_i,\mathbf{z}_j)$ is large.

## D    CONDITIONAL EVIDENCE AS RÉNYI DIVERGENCE BETWEEN POSTERIORS

That Equation (11) pushes posteriors towards each other follows from properties of Rényi divergence with $\alpha\in[0,1]$ and the negative direction of the gradient on the Rényi divergence. In particular, we have (i) that the divergence is a similarity measure of distributions for $\alpha\geq0$ so $D_{\alpha=0}[q||p]=0\implies q=p$, (ii) that $D_\alpha[q\parallel p]$ is everywhere positive, and (iii) the divergence is jointly convex (i.e. convex in both distributions) (Van Erven & Harremos, 2014). Putting it all together, we see from (ii) and (iii) that the extremum at $D_{\alpha=0}[q||p]=0$ is global and from the (negative) gradient we are minimizing $D_{\alpha=0}[q||p]$. So, we have that $-\nabla_{\boldsymbol{\phi}}D_{\alpha=0}[q(\mathbf{z}|\mathcal{D},\boldsymbol{\phi})\parallel p(\mathbf{z}|\mathcal{D})]=0\implies D_{\alpha=0}[q(\mathbf{z}|\mathcal{D},\boldsymbol{\phi})\parallel p(\mathbf{z}|\mathcal{D})]=0$ which from (i) means $q(\mathbf{z}|\mathcal{D},\boldsymbol{\phi})=p(\mathbf{z}|\mathcal{D})$.

## E    ALTERNATIVE ELBO-WITHIN-STEIN DERIVATION

For $\alpha=1$ we can derive the attractive force of $S_\Phi^H$ directly by applying Jensen's inequality to the log conditional evidence, resulting in

$$\nabla_{\boldsymbol{\phi}}\log\mathbb{E}_q\left[\frac{p(\mathcal{D},\mathbf{z}|\boldsymbol{\phi})}{q(\mathbf{z}|\mathcal{D},\boldsymbol{\phi})}\right]\geq\nabla_{\boldsymbol{\phi}}\mathbb{E}_q\left[\log\frac{p(\mathcal{D},\mathbf{z}|\boldsymbol{\phi})}{q(\mathbf{z}|\mathcal{D},\boldsymbol{\phi})}\right]. \tag{18}$$

In ELBO-within-Stein, the attractive force takes the simple form

$$S_\Phi^{\mathrm{ELBO+}}(\boldsymbol{\phi}_i)=\mathbb{E}_{\boldsymbol{\phi}\sim q_\Phi}\left[k(\boldsymbol{\phi}_i,\boldsymbol{\phi})\nabla_{\boldsymbol{\phi}}\mathbb{E}_{q(\mathbf{z}|\boldsymbol{\phi})}\left[\log p(\mathcal{D},\mathbf{z}|\boldsymbol{\phi})\right]-k(\boldsymbol{\phi}_i,\boldsymbol{\phi})\nabla_{\boldsymbol{\phi}}\mathbb{E}_{q(\mathbf{z}|\boldsymbol{\phi})}\left[q(\mathbf{z}|\mathcal{D},\boldsymbol{\phi})\right]\right], \tag{19}$$

and the repulsive force is given by Equation (5).

Table 4: Summary statistics of datasets from the UCI regression benchmark.

| Dataset | Data points | Feature count |
|---|---|---|
| Boston (Harrison Jr & Rubinfeld, 1978) | 506 | 13 |
| Concrete (Yeh, 1998) | 1030 | 8 |
| Energy (Tsanas & Xifara, 2012) | 768 | 8 |
| Power (Tüfekci, 2014) | 9568 | 4 |
| Protein (Rana, 2013) | 45730 | 9 |
| Year (Bertin-Mahieux et al., 2011) | 515345 | 90 |

Table 5: Variational autoencoder architecture for MNIST and OMNIGLOT. **s** denotes a stochastic layer and **d** denotes a deterministic layer. Read the networks left-to-right for guide description and right-to-left for model description.

| Dataset | Architecture | Activation |
|---|---|---|
| MNIST | **d**200-**d**200-**s**50 | `tanh` |
| OMNIGLOT | **d**200-**d**200-**s**100-**d**100-**d**100-**s**50 | `tanh` |

## F    ILLUSTRATING STEIN MIXTURES

To illustrate the use of a SM, consider the variational autoencoder (VAE) (Kingma & Welling, 2013). The VAE simultaneously trains a generative model $p(\mathcal{D}|g_{\boldsymbol{\theta}}(\mathbf{z}))p(\mathbf{z})$ and a variational approximation $q(\mathbf{z}|f_{\boldsymbol{\psi}}(\mathcal{D}))$ of the posterior $p(\mathbf{z}|\mathcal{D})$. Here, $\boldsymbol{\theta}$ and $\boldsymbol{\psi}$ are parameters of the generative neural network $g_{\boldsymbol{\theta}}(\cdot)$ and the inference network $f_{\boldsymbol{\psi}}(\cdot)$, respectively. VAE training is typically done by stochastic variational inference (SVI) (Hoffman et al., 2013) which optimizes $\boldsymbol{\theta}$ and $\boldsymbol{\psi}$ to minimize the ELBO. With a SM, the generative model remains the same, that is, we obtain a point estimate of $\boldsymbol{\theta}$. However, the marginal posterior approximation changes to $1/|\Phi| \sum_{\boldsymbol{\phi} \in \Phi} q(\mathbf{z}|f_{\boldsymbol{\phi}}(\mathcal{D}))$. So with a Stein mixture, each particle $\boldsymbol{\phi}$ parameterizes a separate inference network, i.e. $f_{\boldsymbol{\phi}}(\cdot)$, meaning the guide becomes amortized similar to Shu et al. (2018).

## G    UCI BENCHMARK DETAILS

We compare ELBO-within-Stein for $\alpha \in \{0, 0.5, 1\}$ on BNNs regression point mass (Dirac delta) guide and a RBF kernel. With ELBO-within-Stein we recover a variant of SVGD with a VR gradient rather than the score function. Like Liu & Wang (2016), we use a BNN with one hidden layer of size fifty and a `RELU` activation. We put a Gamma$(1, 0.1)$ prior on the precision of the neurons and the likelihood. For both versions we use 5 particles and update Year for 40 epochs, Protein for 100 epochs and 500 epochs for the rest. We use Adagrad (Duchi et al., 2011) with a step size of 0.05 and a subsample size of 100. All measurements are repeated five times and obtained on a GPU[3].

## H    VAE DETAILS

Following Li & Turner (2016); Burda et al. (2015); Rainforth et al. (2018), we use VAEs with multiple stochastic layers. The idea is to define the model through ancestral sampling as

$$p(\mathbf{x}|\boldsymbol{\theta}) = \sum_{\mathbf{z}_1, \ldots, \mathbf{z}_L} p(\mathbf{z}_L)p(\mathbf{z}_{L-1}|f_{\boldsymbol{\theta}_{L-1}}(\mathbf{z}_L)) \ldots p(\mathbf{x}|f_{\boldsymbol{\theta}_0}(\mathbf{z}_1)),$$

where $\mathbf{x}$ is a data-point (which we will also denote $\mathbf{z}_0$), $\mathbf{z}_1, \ldots, \mathbf{z}_L$ are the $L$ stochastic layers, and $\boldsymbol{\theta}_l$ parameterizes a neural network $f_l$ which takes $\mathbf{z}_{l+1}$ to the parameters of the distribution $p_l$, i.e. $p(\mathbf{z}_l|f_l(\mathbf{z}_{l+1}))$. We then let the guide factor in the opposite direction, resulting in

$$q(\mathbf{z}|\boldsymbol{\phi}, \mathbf{x}) = q(\mathbf{z}_1|f_{\boldsymbol{\phi}_1}(\mathbf{x}))q(\mathbf{z}_2|f_{\boldsymbol{\phi}_2}(\mathbf{z}_1)) \ldots p(\mathbf{z}_L|f_{\boldsymbol{\phi}_L}(\mathbf{z}_{L-1})).$$

---

[3]Quadro RTX 6000 with Cuda V11.4.120

We use the same network architecture as Rainforth et al. (2018) (summarized in Table 5). In Table 5, **s** denotes a stochastic layer and **d** denotes a deterministic layer (affine transforms). We use `tanh` as the activation functions on deterministic layers. Stochastic layers distribute according to a factorized Gaussian distribution, and for the likelihood we use the Bernoulli distribution (hence the binarization of the datasets).

## I   THE EINSTEINVI LIBRARY

We provide a library called `EinSteinVI` for inferring Stein mixtures in the probabilistic programming language (PPL) `NumPyro` (Bingham et al., 2019; Phan et al., 2019). `EinSteinVI` uses $\alpha$-indexed SM inference as its core algorithm as described in Section 3. `NumPyro` is a universal PPL (van de Meent et al., 2018) embedded in Python. `NumPyro` provides specialized constructs for expressing probabilistic models as Python programs and allows executing arbitrary code in its model and guide. The computational backend of `NumPyro` is `Jax` (Frostig et al., 2018), which combines `XLA` (accelerated linear algebra) (Sabne, 2020) program transformations with automatic differentiation. As `EinSteinVI` works with arbitrary guides, `NumPyro` is a well-suited language for embedding `EinSteinVI`. Further, we chose `NumPyro` because:

- `NumPyro` is embedded in Python, the de-facto programming language for data science;
- `NumPyro` includes the necessary data structures for tracking random variables in both model and guide;
- `NumPyro` features stochastic variational inference (SVI) with an application programming interface (API) that is well suited for `EinSteinVI`; and
- `NumPyro` benefits computationally from `Jax`.

### I.1   A EINSTEINVI PROGRAM EXAMPLE

To demonstrate the two modes of VI (SVGD and Stein mixtures) with EinSteinVI, we consider the 1D Gaussian mixture $1/3\mathcal{N}(-2,1) + 2/3\mathcal{N}(2,1)$ (see Figure 3 and Figure 4). The Gaussian mixture is bi-modal and well-suited for the nonparametric nature of SVGD and Stein mixtures. Figure 4 shows that both SVGD[4] and the Stein-mixture naturally capture the bi-modality of the target distribution, compared to SVI with a Gaussian guide. Note the reduction in particles required to estimate the target when using Stein mixtures compared to SVGD. Also, note that the Stein-mixture overestimates the variance and slightly perturbs the locations. The error seen at the right mode for the Stein-mixture with two particles is due to the uniform weighting of the particles in SVGD, and as such is algorithmic. The Stein-mixture will therefore not be able to exactly capture the mixing components of a target mixture model with one particle per component. However, with more particles the mixture can be approximated better as demonstrated using three particles.

### I.2   INTEGRATION WITH NUMPYRO

Integrating `EinSteinVI` with `NumPyro` requires handling transformations between the parameter representation of `NumPyro`[5] and the array representation that ELBO-within-Stein operates on. For this, we rely on `Jax`'s `PyTrees`[6] which converts back and forth between Python dictionaries and array representations.

Algorithm 1 shows the black-box version of $\alpha$-indexed SM inference in `NumPyro`. The algorithm allows SVI to estimate a subset of the parameters and $\alpha$-indexed SM inference the rest. To differentiate the two, we denote parameters updated by SVI with $\psi$ and parameters updated by ELBO-within-Stein with $\phi_i$ (i.e. the Stein particles $\Phi = \{\phi\}_{i=1}^N$). In the model, only SVI can update parameters which we denote by $\theta$. We update $\theta$ and $\psi$ by averaging the loss over the Stein particles. For the Stein particles, the process is more elaborate. First, we convert the set of individual distribution parameters in the guide to a monolithic array using `Jax`'s `PyTrees`. The array represents the particles as a

---

[4]We recover SVGD with a point mass (Delta dirac distribution) on all distributions in the guide.

[5]A dictionary mapping parameters to their values, which can be of arbitrary Python type

[6]`https://jax.readthedocs.io/en/latest/pytrees.html`

```python
def model():
    sample('x', NormalMixture(jnp.array([1 / 3, 2 / 3]),
                              jnp.array([-2.0, 2.0]),
                              jnp.array([1.0, 1.0])))
```

(a) 1D Gaussian mixture model

```python
svi = SVI(
    model,
    AutoNormal(model),
    Adagrad(step_size=1.0),
    Trace_ELBO()
)
results = svi.run(rng_key,
                  num_iterations)
```

```python
stein = SteinVI(
    model,
    AutDelta(model),
    Adagrad(step_size=1.0),
    Trace_ELBO(),
    RBFKernel(),
)
results = stein.run(rng_key,
                    num_iterations)
```

(b) SVI                                   (c) SVGD with EinSteinVI

Figure 3: 1D Gaussian mixture model in NumPyro. We use the deprecated `NormalMixture` over the more general (and more verbose) `MixtureSameFamily` for clarity.

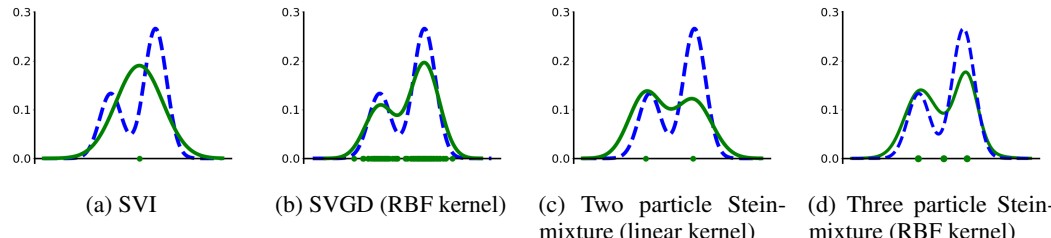

(a) SVI          (b) SVGD (RBF kernel)     (c) Two particle Stein-        (d) Three particle Stein-
                                           mixture (linear kernel)         mixture (RBF kernel)

Figure 4: The blue dashed line is the target pdf, while the solid green line is the density of the particles. We estimate the particle density for SVGD with Gaussian kernel density estimation. We use 100 particles for SVGD, and two or three particles for the Stein-mixture. SVI uses a Gaussian guide.

flattened and stacked `Jax` array. Then, we compute a kernel on the particles, delegated to the kernel interface (see Appendix I.3) as the computation is kernel-dependent.

We apply `Jax`'s `vmap` (Frostig et al., 2018; Phan et al., 2019) operator to compute the Stein forces for each particle in a vectorized manner. As we compute the Stein forces in unconstrained space, we must correct them by the Jacobian of the bijection to constrained space. Naively computing the Jacobian on the monolithic array incurs a massive memory overhead. However, as NumPyro registers a bijection for each distribution parameter, we can eliminate the overhead by computing the Jacobian on the `Jax` representations of the individual distribution parameters rather than the monolith. Like computing the Stein forces, the correction is embarrassingly parallel so that we can use a `vmap` operator again. Inside the `vmap` we nest a `tree_map` to do the appropriate conversion between representations. Finally, we convert the monolithic array to its `NumPyro` representation and return the expected changes for SVI- and Stein- parameters.

---

**Require:** SVI parameters $\boldsymbol{\theta}$ and $\boldsymbol{\psi}$, Stein parameters $\{\boldsymbol{\phi}_i\}_{i=1}^N$, model $p_{\boldsymbol{\phi}}(\mathbf{z}, \mathbf{x})$, guide $q_{\boldsymbol{\theta},\boldsymbol{\psi}}(\mathbf{z})$, loss $\mathcal{L}_\alpha$, kernel interface KI.
**Ensure:** Parameter changes based on SVI ($\Delta\boldsymbol{\theta}$, $\Delta\boldsymbol{\psi}$) and hierarchical Stein forces ($\{\Delta\boldsymbol{\phi}_i\}_{i=1}^N$).

    **procedure** UPDATE($\boldsymbol{\theta}$, $\boldsymbol{\psi}$, $\{\boldsymbol{\phi}_i\}_{i=1}^N$, $p_{\boldsymbol{\theta}}$, $q_{\boldsymbol{\phi},\boldsymbol{\psi}}$)
        $\Delta\boldsymbol{\theta} \leftarrow \mathbb{E}_{\boldsymbol{\theta}}[\nabla_{\boldsymbol{\theta}}\mathcal{L}_\alpha(p_{\boldsymbol{\theta}}, q_{\boldsymbol{\phi},\boldsymbol{\psi}})]$
        $\Delta\boldsymbol{\psi} \leftarrow \mathbb{E}_{\boldsymbol{\psi}}[\nabla_{\boldsymbol{\psi}}\mathcal{L}_\alpha(p_{\boldsymbol{\theta}}, q_{\boldsymbol{\phi},\boldsymbol{\psi}})]$
        $\{\mathbf{a}_i\}_i \leftarrow \text{PyTreeFlatten}(\{\boldsymbol{\phi}_i\}_{i=1}^N)$
        $k \leftarrow \text{KI}(\{\mathbf{a}_i\}_{i=1}^N)$

        **procedure** HSTEIN-FORCES($\mathbf{a}_i$)        ▷ Calculate forces per particle for higher-order vmap function.
            $\boldsymbol{\theta}_i \leftarrow \text{PYTREERESTORE}(\mathbf{a}_i)$
            $\Delta\mathbf{a}_i \leftarrow \sum_{\mathbf{a}_j} k(\mathbf{a}_j, \mathbf{a}_i)\nabla_{\mathbf{a}_i}\mathcal{L}_\alpha(p_{\boldsymbol{\phi}}, q_{\boldsymbol{\theta}_i,\boldsymbol{\psi}}) + \nabla_{\mathbf{a}_i}k(\mathbf{a}_j, \mathbf{a}_i)$
            **return** $\Delta\mathbf{a}_i$
        **end procedure**

        $\{\Delta\mathbf{a}_i\}_i \leftarrow \text{VMap}(\{\mathbf{a}_i\}_i, \text{HSTEIN-FORCES})$
        $\{\Delta\boldsymbol{\phi}_i\}_{i=1}^N \leftarrow \text{PYTREERESTORE}(\{\Delta\mathbf{a}_i\}_{i=1}^N)$
        **return** $\Delta\boldsymbol{\theta}$, $\Delta\boldsymbol{\psi}$, $\{\Delta\boldsymbol{\phi}_i\}_{i=1}^N$
    **end procedure**

**Algorithm 1:** $\alpha$-indexed Stein Mixture inference

---

### I.3 KERNEL INTERFACE

The kernel interface is straightforward. To extend the interface, users must implement the `compute` function, which accepts as input the current set of particles, the mapping between model parameters and particles, and the loss function $\mathcal{L}$ and returns a differentiable kernel $k$. Table 6 gives the complete list of kernels in EinSteinVI.

Table 6: Kernels included in the EinSteinVI library.

| Kernel | Definition | Comments | Type | Reference |
|---|---|---|---|---|
| Radial Basis Function (RBF) | $\exp(\frac{1}{h} \parallel \mathbf{x} - \mathbf{y} \parallel_2^2)$ | | scalar | Liu & Wang (2016) |
| | $\exp(\frac{1}{h}(\mathbf{x} - \mathbf{y}))$ | | vector | Pyro[7] |
| Inverse Multi-Quadratic (IMQ) | $(c^2 + \parallel \mathbf{x} - \mathbf{y} \parallel_2^2)^{\beta}$ | $\beta \in (-1, 0)$ and $c > 0$ | scalar | Gorham & Mackey (2017) |
| Random Feature Expansion | $\mathbb{E}_{\mathbf{w}}[\boldsymbol{\phi}(\mathbf{x}, \mathbf{w})\boldsymbol{\phi}(\mathbf{y}, \mathbf{w})]$ | $\boldsymbol{\phi}(\mathbf{x}, \mathbf{w}) = \sqrt{2}\cos(\frac{1}{h}\mathbf{w}_1^{\top}\mathbf{x} + \mathbf{w}_0)$ where $\mathbf{w}_0 \sim \mathrm{Unif}(0, 2\pi)$ and $\mathbf{w}_1 \sim \mathcal{N}(0, I_d)$ | scalar | Liu & Wang (2018) |
| Linear | $\mathbf{x}^{\top}\mathbf{y} + 1$ | | scalar | Liu & Wang (2018) |
| Mixture | $\sum_i \boldsymbol{\omega}_i k_i(\mathbf{x}, \mathbf{y})$ | $\{k_i\}_i$ individual kernels, weights $\boldsymbol{\omega}_i$ | scalar, vector, matrix | Liu & Wang (2018) |
| Scalar-based Matrix | $k(\mathbf{x}, \mathbf{y})\boldsymbol{I}_d$ | $k$ scalar-valued kernel | matrix | Wang et al. (2019) |
| Vector-based Matrix | $\mathrm{diag}(k(\mathbf{x}, \mathbf{y}))$ | $k$ vector-valued kernel | matrix | Wang et al. (2019) |
| Graphical | $\mathrm{diag}(\{K^{(\ell)}(\mathbf{x}, \mathbf{y})\}_{\ell})$ | $\{K^{(\ell)}\}_{\ell}$ scalar-valued kernels, each for a unique partition of latent variables | matrix, placed on the diagonal | Wang et al. (2019) |
| Constant Pre-conditioned | $\boldsymbol{Q}^{-\frac{1}{2}} K(\boldsymbol{Q}^{\frac{1}{2}}\mathbf{x}, \boldsymbol{Q}^{\frac{1}{2}}\mathbf{y})\boldsymbol{Q}^{-\frac{1}{2}}$ | $K$ is an inner matrix-valued kernel and $\boldsymbol{Q}$ is a preconditioning matrix like the Hessian $-\nabla_{\bar{\mathbf{z}}}^2 \log p(\bar{\mathbf{z}}|\mathbf{x})$ or Fisher information $-\mathbb{E}_{\mathbf{z} \sim q_{\mathcal{Z}}(\mathbf{z})}[\nabla_{\mathbf{z}}^2 \log p(\mathbf{z}|\mathbf{x})]$ matrices | matrix | Wang et al. (2019) |
| Anchor Point Preconditioned | $\sum_{\ell=1}^{m} K_{\boldsymbol{Q}_{\ell}}(\mathbf{x}, \mathbf{y})\boldsymbol{\omega}_{\ell}(\mathbf{x})\boldsymbol{\omega}_{\ell}(\mathbf{y})$ | $\{\mathbf{a}_{\ell}\}_{\ell=1}^{m}$ is a set of anchor points, $\boldsymbol{Q}_{\ell} = \boldsymbol{Q}(\mathbf{a}_{\ell})$ is a preconditioning matrix for each anchor point, $K_{\boldsymbol{Q}_{\ell}}$ is an inner kernel conditioned using $\boldsymbol{Q}_{\ell}$, and $\boldsymbol{\omega}_{\ell}(\mathbf{x}) = \mathrm{softmax}_{\ell}(\{\mathcal{N}(\mathbf{x}|\mathbf{a}_{\ell'}, \boldsymbol{Q}_{\ell'}^{-1})\}_{\ell'})$ | matrix | Wang et al. (2019) |

