# OpenReview forum: "ELBO-ing Stein Mixtures"
_ICLR.cc/2023/Conference — Submitted to ICLR 2023_

### Official Review · Reviewer_Ra8d · 2022-10-17

**Confidence:** 2
**Clarity, Quality, Novelty And Reproducibility:** The paper is well written, mostly cle…
**Correctness:** 2
**Technical Novelty And Significance:** 2
**Empirical Novelty And Significance:** 2
**Recommendation:** 3

**Strength And Weaknesses:**

Strength:

1. This paper is overall well-written and the literature review is thorough.

Weaknesses:

1. The motivation of the proposed method is vague. No comparison to other SVGD variants.
2. The orginality is low as the major contribution seems to be some empirical investigation of a hyperparameter.
3. The paper is a little bit hard to follow. What is the main difference between stein mixture and a simple mixture model?

**Summary Of The Paper:**

This paper investigate a pevious work on stein mixtures and show that it corresponds to inference with the Renyi $\alpha$-divergence for $\alpha=0$, and using other values of $\alpha$ may stablize the inference.

**Summary Of The Review:**

My main concern is that the original contribution of this work is not significant enough.

---

> ### Author Response · Authors · 2022-11-16
> **Author rebuttal**
>
> ### Motivation and significance
> Please see the [Motivation and significance comment](https://openreview.net/forum?id=gRgCyyYBR4o&noteId=tapkzE3Lki).
>
> ### No comparison to other SVGD variants.
> Please see the section [Empirical demonstration of improvement relative to existing methods](https://openreview.net/forum?id=gRgCyyYBR4o&noteId=1CTLSrLRzr) in rebuttal to the second reviewer 2 (r9He)
>
> ### The motivation of the proposed method is vague.
> Please see [Although more empirical results can make the paper much stronger](https://openreview.net/forum?id=gRgCyyYBR4o&noteId=3DWV-6c_8DA) in the rebuttal for the first reviewer (k1hY).
>
> ### The paper is a little bit hard to follow. What is the main difference between a Stein mixture and a simple mixture model?
> When interpreted in terms of a probabilistic model, a Stein mixture is simply a restricted mixture model, i.e., a mixture model with n components where all mixture weights are equal to $\frac{1}{n}$. In terms of variational inference, using a Stein mixture approximation thus corresponds to using a restricted mixture approximation to any (differentiable) model.
>
> Stein mixture inference uses the relationship between the gradient of the KL of a push-forward measure of a set of particles and the posterior (see [Wang & Liu (2017)](https://arxiv.org/abs/1608.04471)). The value of this new approach lies in avoiding the computation of an intractable entropy factor. The alternative is to approximate it, which introduces a bias (see [Ranganath et. al (2016)](https://arxiv.org/abs/1511.02386)).

---

### Official Review · Reviewer_r9He · 2022-10-24

**Confidence:** 4
**Correctness:** 2
**Technical Novelty And Significance:** 3
**Empirical Novelty And Significance:** 2
**Recommendation:** 3

**Clarity, Quality, Novelty And Reproducibility:**

* I found the paper mostly well written but had trouble following it in some places.  For example, around equation 2 it is very unclear whether the $phi_j$ are part of the model or if they are hyper-parameters of the variational approximation since they are written to have a density under p.  This should be made much more explicit, as the notation in equation 2 suggests that $phi$ exists in the original model $p$.
* The work is original.

**Strength And Weaknesses:**

Strengths
* The explanation and demonstration of the trade-off between the need for a large number of Monte Carlo samples when estimating the gradients without too much bias / variance is nicely laid out and is clear.
* The introduction of alpha-divergences into Stein mixtures is a nice application of a compelling line of work on generalized variational inference and I was glad to see it explored here.
* The worked out toy example with the analytic fixed points is very nice and illuminating!

Weaknesses:
* The authors should better defend a key, undefended premise of their work about which I was unconvinced: that Stein mixtures alleviate an “exponential” dependence of SVGD on the dimensionality on the model.  This premise is not made precise nor is it attributed to previous work that establishes it more precisely.  Why is an exponential number of particles needed for SVGD?  This is certainly not the case for MCMC for example.
* If there is a situation where the dependence is exponential with dimension, would the authors please explain (ideally with a theoretical result) such a situation?
* Or if there is no theoretical justification or proof to be made about the better dependence on dimension, empirical validation of this claim is needed.  For example, it would help to have an illustrative example of a case (e.g. in simulation) where the dimension scaling is exponential with the dimension. E.g. perhaps a plot with an increasing dimension on the x axis and the number of particles needed to achieve some level of posterior precision on the y-axis (for all three of the SVGD method, Stein mixtures, and the new proposed method).
* Or alternatively, if this exponential dependence on dimension is known from prior theory or empirics, even replicating those prior results (with citation) would strengthen the paper.
* Or if the dependence is not actually exponential, the authors must either be precise about the way in which dimensionality poses a problem that is addressed by their approach or drop this as the premise of the method.

* Given that the paper is about an improvement upon Nalisnick and Smyth 2017 (whose applications and limitations most readers may not be very familiar with), the paper could benefit from more justification as to why this method is important/useful and why it must be improved.  At present, I found it challenging to see what problem the new methodology was trying to solve that Nalisnick and Smyth 2017 had not solved.
* On problems where the developed methodology is applicable, how does it compare to other Bayesian inference methods?  E.g. HMC, MFVB, Laplace approximations etc., in terms of computational cost and accuracy (experiments are not necessarily needed, but I think commentary at least is).  For the UCI comparisons, previously run baselines could be used.

**Summary Of The Paper:**

The authors generalized the Stein Mixtures approach of Nalisnick and Smyth 2017 to general alpha-divergences.  The paper is mostly nicely written and was a good read.  But it did not have a clear motivation nor impressive empirics.

**Summary Of The Review:**

The paper is an enjoyable read about extending Stein mixtures to use alpha divergences.  But I recommend rejection because in my assessment it falls short on:
* Motivating how or why the proposed method addresses limitations of existing methods (particularly dimension dependence) and
* Empirical demonstration of improvement relative to existing methods.

---

> ### Author Response · Authors · 2022-11-16
> **Author rebuttal**
>
> ### Motivation and significance
> Please see the [Motivation and significance comment](https://openreview.net/forum?id=gRgCyyYBR4o&noteId=tapkzE3Lki).
>
> ### SVGD and the curse of dimensionality
> We agree that this topic deserves a more precise discussion. The curse of dimensionality for SVGD was first explored in [Wang et al. (2018)]( https://arxiv.org/abs/1711.07168), demonstrating the problem empirically for a high dimensional multivariate Gaussian distribution. The same problem is studied in further detail in [Ba et al. (2022)](https://openreview.net/pdf?id=Qycd9j5Qp9J]), which provides a theoretical analysis of the Gaussian case mentioned above. [Ba et al. (2022)](https://openreview.net/pdf?id=Qycd9j5Qp9J]) shows that variance estimation scales inversely with dimensionality for SVGD. We have made the statement concerning the curse of dimensionality in the abstract more precise and added a paragraph to our revision that discusses the results mentioned above.
>
>
> ### The improvement over Nalisnick & Smyth 2017
>
> The core issue is that SVGD suffers from variance collapse when the dimensionality of the model increases (see [Ba et al. (2022)](https://openreview.net/pdf?id=Qycd9j5Qp9J])). In practice, we need more particles to (correctly) estimate the variance of higher dimensional models. Nalisnick & Smyth (2017) suggest to let each particle represent a neighborhood of particles in order to address this issue. In their approach, each particle parameterizes a density, resulting in an approximating distribution corresponding to a mixture model.
> We extend and generalize their work by connecting their inference algorithm with the $\alpha$-Renyi divergence and demonstrating that $\alpha=1$ leads to better results empirically. This is demonstrated on a wide range of benchmarks (see Tables 2, 3 and 4).
>
> In addition, we hypothesize and provide convincing numerical evidence that the reason for the improvement is that the change in the order of the Renyi divergence improves the signal-to-noise ratio of the gradient estimation. This stabilizes the convergence of the inference. Hence, we also provide considerable insight into why our generalization of Nalisnick & Smyth (2017) performs better.
>
> ### Empirical demonstration of improvement relative to existing methods.
> Please see the section [Although more empirical results can make the paper much stronger](https://openreview.net/forum?id=gRgCyyYBR4o&noteId=3DWV-6c_8DA) in the rebuttal for the first reviewer (k1hY).

---

### Official Review · Reviewer_k1hY · 2022-10-25

**Confidence:** 3
**Correctness:** 3
**Technical Novelty And Significance:** 3
**Empirical Novelty And Significance:** 3
**Recommendation:** 8

**Clarity, Quality, Novelty And Reproducibility:**

Some minor comments in terms of clarity:

- the third contribution mentionds global latent and local latent variables, could the authors clarify their definition. The explanation in section 3.1 is not crystal clear to the reviewer.

- on page 2, when the authors claim the motivation of the method, they state "for such estimates, we cannot expect XXX as stein force will compensate the gradient estimation error by a counter force", can the authors be more specifc?


**Strength And Weaknesses:**

Strengths:

- The paper is overall well-written and the method is clearly explained.  The reviewer really like the writing in section 2 that builds up the knowledge step by step
- The literature review is thorough
- The connection with the Renyi divergence and the VR bound seems interesting

Weakness:

- Although the connection with existing works is interesting, the reviewer is not fully convinced why ELBO-within-Stein is preferred over the existing ones.

- given a real data analysis, how should one pick the value of $\alpha$



**Summary Of The Paper:**

The paper aims to improve the numerical stability of stein mixture in variational inference. To achieve this, the proposed EBLO-within-stein method allows the choice of a class of hierarchical attractives forces indexed by the hyper-parameter $\alpha$ in the Renyi divergence. Different values of $\alpha$ acounts for different types of latent variables, which could be measured by the signal-to-noise ratios. The paper includes empirical results to show various effects of $\alpha$ on real data analysis.

**Summary Of The Review:**

Although more empirical results can make the paper much stronger, this is an overall well-executed paper. The authors need clarify on a few raised questions.

---

> ### Author Response · Authors · 2022-11-16
> **Author rebuttal**
>
> ### Motivation and significance
> Please see the [Motivation and significance comment](https://openreview.net/forum?id=gRgCyyYBR4o&noteId=tapkzE3Lki).
>
> ### Difference between local- and global-latent models
> A local latent model has a latent random variable z associated with each data point. For an I.I.D. dataset $D=\\{x_i\\}_{i=1}^N$, the joint distribution $p(D, Z) = \prod_i p(x_i | z_i)p(z_i)$.
>
> A global latent model shares the latent variable across data points. In this case, $p(D, Z) = \prod_i p(x_i | z)p(z)$. We clarified this in the revision.
>
> ### Given a real data analysis, how should one pick the value of \alpha?
> As $\alpha$-indexed Stein mixtures are implemented as an inference engine in NumPyro, a hyperparameter search would be fairly simple for practical applications. However, our experimental evidence indicates $\alpha=1$ is a good default.
>
> ### Inference convergence and noisy gradients
> We cannot expect the inference to converge when using noisy gradient estimations because the Stein force moves in the steepest KL descent direction. Each iteration will adjust for the error in the gradient direction from the previous step; as the gradient estimation is noisy (has high relative variance), the Stein force will never become zero. Thus, the inference method will fail to converge properly. This is why we suggest using $\alpha>0$, as it provides a better SN-ratio (a measure of relative variance).
>
> ## Although more empirical results can make the paper much stronger
> We have extended the BNN experiments with results from MFVI, the Laplace approximation, and SVGD to demonstrate ELBO-within-Stein performance better. The added results demonstrate that EoS achieves lower (better) RMSE on all BNN benchmarks. Note that Eos outperforms SVGD even while using fewer particles. This demonstrates the advantage of letting the particles parameterize a guide. On the log-likelihood evaluation for BNNs, EoS outperforms Nalisnick & Smith (2017); as both methods are mixture approximations, the better goodness-of-fit for the BNN with $\alpha$=1 demonstrates that EoS yields better approximations.

---

### Author Response · Authors · 2022-11-16
**Paper revision changelog**

- Added local latent and global latent models distinction
- The corrected growth in particles with model dimensionality
- Moved UCI description table to the Appendix
- Moved delta results to the main article
- Added MFVI, SVGD, and Laplace approximation results to BNN example (Table 2 and 3)
- Added paragraph about the curse of dimensionality

---

### Author Response · Authors · 2022-11-18
**Motivation and significance**

Stein variational gradient descent (SVGD) requires an increasing amount of particles to represent higher dimensional models ([Ba et al. 2022](https://openreview.net/pdf?id=Qycd9j5Qp9J])). This is computationally burdensome, as SVGD scales quadratically in complexity with the number of particles. Nalisnick & Smythe (2017) empirically demonstrate that __we can improve performance on real-world (high-dimensional) problems while using fewer particles, by lifting particles to densities__. We generalize the method of Nalisnick & Smythe (2017) using the Renyi $\alpha$-divergence, and prove that their method corresponds to using $\alpha=0$. This is important, as we show that __using $\alpha=1$ instead of $\alpha=0$ leads to significantly better results__. We show this for a wide range of problems, ranging from regression on housing prices to image classification, using Bayesian neural networks and variational autoencoders. We also show that our method does better than other methods, including MFVI, Laplace approximation, and SVGD.  In addition, we demonstrate __why our method does better__: the improvement can be attributed to reducing the noise in the gradient estimation. Finally, we provide a ready-to-use implementation of our method as part of the deep probabilistic programming language Numpyro.

---

### Decision · Program_Chairs · 2023-01-20

**Decision:**

Reject

**Justification For Why Not Higher Score:**

NA

**Justification For Why Not Lower Score:**

NA

**Metareview: Summary, Strengths And Weaknesses:**

This work considers an extension of Stein variational gradient descent to the the case where the training objective captures an f-divergence between the proposal and target distribution, and develops algorithmic machinery and convergence guarantees for such methods.

The strengths of this work are in the writing quality, technical precision, and numerical validation of the proposed approach.

The weaknesses are in establishing the proposed generalization addresses a key shortcoming of prior work. In particular, both theoretically and numerically, it is not clear why the proposed generalization can address a gap in prior works, and fails to illuminate the practical significance of the additional generality. This is a key sticking point among the reviewers with which I tend to agree. For this reason, the paper is below the bar of acceptance.